# Simulating the quantum switch with quantum circuits is computationally hard

**Jessica Bavaresco** [1,2]**, Hlér Kristjánsson** [3,4,5,6]**, Mio Murao** [5,7]**, Tatsuki Odake** [5]**,
Marco Túlio Quintino** [2] ✉**, Philip Taranto** [5,8] **& Satoshi Yoshida** [5]

Higher-order transformations acting on input quantum channels in an indefinite causal order—such as the quantum switch—cannot be described by quantum circuits using the same number of calls to the input channels. A natural question is whether they can be simulated, i.e., whether their action can be exactly and deterministically reproduced by a quantum circuit with more calls to the input channels. Here, we prove that the quantum switch acting on two $n$-qubit channels cannot be simulated by any quantum circuit using $k$ calls to one channel and one to the other, if $k < 2^n$. This establishes an exponential separation in quantum query complexity between processes with indefinite causal order and quantum circuits. Moreover, even with one extra call to both input channels, such a simulation remains impossible. We further demonstrate the robustness of this separation by extending the result to probabilistic and approximate simulations scenarios.

Indefinite causal order is a fundamental property that emerges in the exploration of higher-order transformations within quantum theory[1–5]. Higher-order quantum transformations are operations whose inputs and outputs are not quantum states, but rather transformations that themselves act on quantum states, such as quantum channels. An example of such a transformation is a quantum circuit with open slots where arbitrary quantum channels can be inserted and acted upon in a sequential, temporally ordered manner[1,3]. Remarkably, there also exist well-defined higher-order transformations that act on their input quantum channels in an indefinite causal order[6–8]. Such transformations cannot be described by any quantum circuit that uses the same number of calls, or queries, of the input quantum channels, challenging the standard notion of computation in which operations are performed on a system in a fixed order.

The ability to perform operations in such an indefinite order has been shown to provide advantages in a variety of information-processing settings, such as quantum channel discrimination[9–11], quantum metrology[12–15], quantum computational complexity[16,17], quantum query complexity[18], transformations of black-box unitaries and isometries[19–23], among others.

A prominent example of indefinite causal order that is responsible for several of these theoretically predicted advantages is the quantum switch[6], a higher-order transformation that coherently controls the causal order of two quantum channels. The information-processing advantages of the quantum switch have mostly been shown via comparison with higher-order transformations that act in a fixed order on a single call of each input channel[8–10,12]. However, their true practical significance hinges on the extent to which these advantages would still hold when comparing the quantum switch with quantum circuits that use a larger number of calls to its inputs.

In the context of quantum computation, whether the quantum switch exhibits a true complexity-theoretic advantage depends upon whether its action can be efficiently reproduced by using quantum circuits, given extra queries to the input channels. Until now, no exponential separation has been demonstrated between the query complexity of computations using indefinite causal order versus quantum circuits. In fact, for unitary channels, the action of the

[1]Department of Applied Physics, University of Geneva, Geneva, Switzerland. [2]Sorbonne Université, CNRS, LIP6, Paris, France. [3]Perimeter Institute for Theoretical Physics, Waterloo, ON, Canada. [4]Institute for Quantum Computing, University of Waterloo, Waterloo, ON, Canada. [5]Department of Physics, Graduate School of Science, The University of Tokyo, Bunkyo-ku, Tokyo, Japan. [6]Department of Computer Science and Operations Research, Université de Montréal, Montréal, QC, Canada. [7]Trans-scale Quantum Science Institute, The University of Tokyo, Bunkyo-ku, Tokyo, Japan. [8]Department of Physics & Astronomy, University of Manchester, Manchester, UK. ✉e-mail: Marco.Quintino@lip6.fr

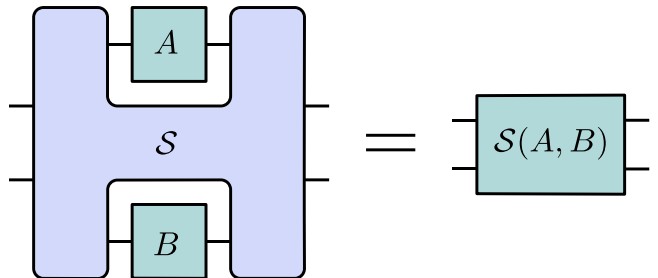

**Fig. 1 | The quantum switch transformation.** The quantum switch $\mathcal{S}$ is a higher-order transformation that takes as input any two quantum channels $A$ and $B$ and transforms them into a another quantum channel $\mathcal{S}(A,B)$. The resulting channel $\mathcal{S}(A,B)$ acts on a qubit control system and a qudit target system.

quantum switch can be reproduced by a quantum circuit with just one extra query[6], significantly limiting the computational power of the quantum switch in this case. However, a crucial open question is whether this limitation extends to general quantum channels.

More concretely, this question can be phrased in terms of the simulability of the quantum switch, or more generally, of any higher-order transformation with indefinite causal order:

> Can a higher-order transformation with indefinite causal order that acts on arbitrary quantum channels be simulated by quantum circuits that have access to more calls of the input quantum channels?

In the case where the quantum circuit performing the simulation has access to an infinite number of calls of the input channels, the answer is: yes. In this case, a process tomography protocol[3,24,25] can completely characterize the input channels and simply prepare the output channel expected from the higher-order transformation with indefinite causal order. However, such a simulation requires infinite resources. Another possible way that the quantum switch can be simulated is using post-selection[6,26]. However, such a simulation does not work deterministically. Additionally, in the particular case where the input channels are restricted to being unitary channels, it has been shown that a simulation of the quantum switch exists[6]. However, such a simulation is not universal, inasmuch as it only works for a restricted set of input channels.

We hence ask the question, which has remained largely unexplored so far: does any finite number of extra calls of the input channels suffice to deterministically and universally simulate the action of the quantum switch with a quantum circuit?

In this work, we prove that the quantum switch acting on arbitrary channels $A$ and $B$ cannot be deterministically and universally simulated by a quantum circuit (or even a quantum circuit with classical control of the causal order[27]) that has access to $k_A < 2^n$ calls of $A$ and $k_B = 1$ call of $B$, where $A$ and $B$ are arbitrary $n$-qubit quantum channels. Our theorem demonstrates an exponential separation in quantum query complexity for computational tasks using quantum processes with indefinite causal order versus quantum circuits with fixed or classically-controlled causal order, in terms of the number of qubits. We moreover prove that when one extra call of each quantum channel is available ($k_A = k_B = 2$), it remains impossible to simulate the action of the quantum switch, even for single-qubit channels. We demonstrate the extent of the robustness of these results in two different ways: with probabilistic and approximate simulations. We show that even when approximate simulations are considered, the probability of simulating a higher-order operation which is $\epsilon$-close to the quantum switch is significantly below one for an $\epsilon$ significantly above zero. We furthermore thoroughly analyze the problem of simulating the quantum switch when it acts only on part of its input quantum channels or on

quantum instruments. Finally, we show some new particular (non-universal) cases in which simulations of the quantum switch are possible. In light of these results, we conjecture that a deterministic simulation of the quantum switch, if possible, would require a number of queries to both quantum channels that grows at least exponentially with the size of the system they act upon. If our conjecture holds true, it would imply that processes with indefinite causal order cannot be efficiently simulated even by quantum circuits with classically-controlled causal order.

## Results
### Quantum switch simulation

The quantum switch $\mathcal{S}$ is a higher-order transformation that takes two arbitrary quantum channels (i.e., completely positive, trace-preserving maps) $A$ and $B$ as input, where $A : \mathcal{L}(\mathcal{H}^{A_I}) \rightarrow \mathcal{L}(\mathcal{H}^{A_O})$ and $B : \mathcal{L}(\mathcal{H}^{B_I}) \rightarrow \mathcal{L}(\mathcal{H}^{B_O})$ are channels that act on qudit systems, and transforms them into a channel $\mathcal{S}(A,B) : \mathcal{L}(\mathcal{H}^{c_I} \otimes \mathcal{H}^{t_I}) \rightarrow \mathcal{L}(\mathcal{H}^{c_O} \otimes \mathcal{H}^{t_O})$ that acts on a qubit control system and a qudit target system. The output channel resulting from the action of the quantum switch[6] on its input channels is defined as

$$\mathcal{S}(A,B)[\sigma_c \otimes \rho_t] := \sum_{i,j} \mathsf{S}_{ij}(\sigma_c \otimes \rho_t)\mathsf{S}_{ij}^\dagger, \tag{1}$$

where $\sigma_c \in \mathcal{L}(\mathcal{H}^{c_I})$ is the state of the input qubit control system, $\rho_t \in \mathcal{L}(\mathcal{H}^{t_I})$ is that of the qudit target system, and $\mathsf{S}_{ij}$ is given by

$$\mathsf{S}_{ij} := |0\rangle\langle 0| \otimes \mathsf{B}_j\mathsf{A}_i + |1\rangle\langle 1| \otimes \mathsf{A}_i\mathsf{B}_j, \tag{2}$$

where $\mathsf{A}_i : \mathcal{H}^{A_I} \rightarrow \mathcal{H}^{A_O}$ and $\mathsf{B}_j : \mathcal{H}^{B_I} \rightarrow \mathcal{H}^{B_O}$ are Kraus operators[28] of the channels $A$ and $B$, respectively; i.e., $A[\rho] = \sum_i \mathsf{A}_i \rho \mathsf{A}_i^\dagger$ and $B[\rho] = \sum_i \mathsf{B}_i \rho \mathsf{B}_i^\dagger$. This transformation is depicted in Fig. 1. The quantum switch acts on its input channels in an order that is conditioned on the state of a quantum control system. Since the quantum control system may be initiated in a superposition state, such as $|+\rangle := \frac{1}{\sqrt{2}}(|0\rangle + |1\rangle)$, the overall quantum switch transformation may be understood as a conditioned superposition of two different circuits, one with the control system in state $|0\rangle$ and the quantum channels being applied in the order $A$ before $B$, and another with the control system in state $|1\rangle$ and the quantum channels being applied in the order $B$ before $A$.

A deterministic and exact simulation of the quantum switch is a higher-order transformation $\mathcal{C}$ that obeys causal constraints and that acts on a finite number $k_A$ and $k_B$ of calls to the channels $A$ and $B$, respectively, in such a way that the channel $\mathcal{C}(A^{\otimes k_A}, B^{\otimes k_B}) : \mathcal{L}(\mathcal{H}^{c_I} \otimes \mathcal{H}^{t_I}) \rightarrow \mathcal{L}(\mathcal{H}^{c_O} \otimes \mathcal{H}^{t_O})$ resulting from this transformation satisfies

$$\mathcal{C}(A^{\otimes k_A}, B^{\otimes k_B}) = \mathcal{S}(A,B) \quad \forall A, B, \tag{3}$$

where $A, B$ are arbitrary quantum channels. Natural causal constraints that one might impose on the simulation is to require that $\mathcal{C}$ be described by a fixed-order quantum circuit with open slots, called a quantum comb[1,3]. A more general strategy for the simulation—which could nevertheless be interpreted as having a definite causal order—would be to impose that $\mathcal{C}$ is described by open-slot quantum circuits that have classical control of causal order. This class of higher-order transformations, proposed in ref. 27 and called "QC-CCs", is larger than the set of quantum combs, allowing for classical mixtures of open-slot quantum circuits as well as for classically-controlled dynamical causal orders. Hence, permitting this class gives more power to the simulation as compared to quantum combs, while still allowing for a causally ordered interpretation of the simulation (see Supplementary Note 1 in the Supplementary Information file for a formal definition). We furthermore require the simulation to be universal: the same simulation $\mathcal{C}$ must work for all input pairs of quantum channels. See Fig. 2 for a graphical representation of Eq. (3). The question of

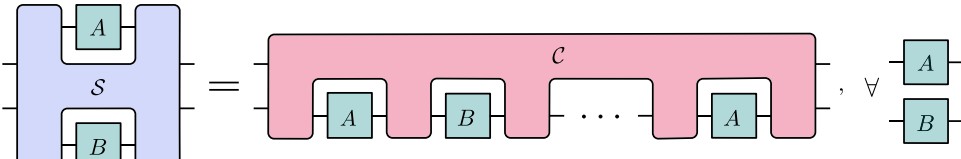

**Fig. 2 | Simulating the quantum switch.** A higher-order transformation $\mathcal{C}$ corresponding to an open-slot quantum circuit with fixed or classically-controlled causal order, which acts on several copies of the input quantum channels $A$ and $B$, is a simulation of the quantum switch $\mathcal{S}$ if it reproduces the action of the quantum switch on all arbitrary channels $A$ and $B$.

simulability then boils down to whether there exists, for some finite number of calls $k_A$ and $k_B$, a simulation $\mathcal{C}$ that obeys causal constraints and that satisfies Eq. (3).

This notion of computation, where the inputs and outputs are quantum rather than classical, is suitable to treat inherently quantum problems and has been employed beyond the higher-order quantum computing paradigm explored here in problems ranging from the SWAP test[29,30] to Hamiltonian simulations[31,32] and quantum property testing[33].

### Go theorem: an explicit non-universal simulation

For a particular case of input channels, it is known that a non-universal simulation of the quantum switch exists. As first shown in the paper that originally defined the quantum switch[6], in the particular case where the input channels are unitary (i.e., reversible), a simulation of the quantum switch by a quantum circuit is possible, requiring only an extra use of one of the input channels.

The simulation presented in ref. 6 is given by the quantum circuit

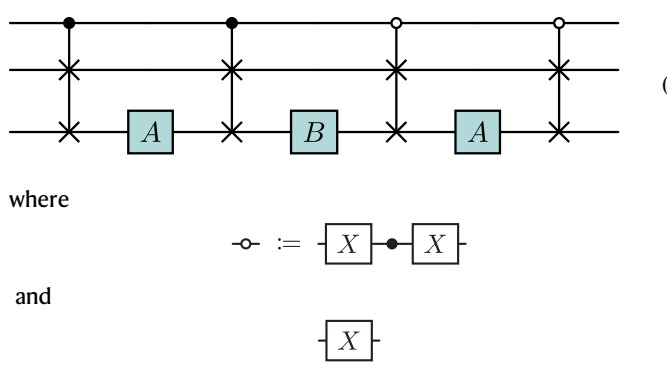

$$\tag{4}$$

where

$$\multimap \hspace{-2pt}\circ := \boxed{X} \!-\!\bullet\!-\! \boxed{X}$$

and

$$\boxed{X}$$

is the NOT gate. Here, the first circuit line corresponds to the control qubit system, the second to an auxiliary system, and the third to the target qudit system. This circuit can equivalently be represented by the Kraus operators $\{C_{iji'}\}_{iji'}$, where

$$C_{iji'} := |0\rangle\langle 0| \otimes A_{i'} \otimes B_j A_i + |1\rangle\langle 1| \otimes A_i \otimes A_{i'} B_j. \tag{5}$$

Since, in general $i \neq i'$, the transformation acting on the auxiliary system does not typically factor out from the transformation acting on the qubit and control systems. By comparison with Eq. (2), it is straightforward to see that the resulting transformation does not simulate the quantum switch for arbitrary quantum channels. However, in the case where $A$ and $B$ are unitary channels, and hence are described by a single Kraus operator (i.e., $i = i' = j = 0$), the transformation on the auxiliary system factorizes from that on the control and target systems. In this particular case, it is straightforward to see that, for any input state of the auxiliary system, when the output auxiliary system is discarded, the quantum circuit in Eq. (4) performs a (non-universal) simulation of the quantum switch.

A crucial point to note is that the circuit in Eq. (4) acts on the input channels in the order "ABA". If one were to change the order of the input channels to either "AAB" or "BAA", this circuit no longer

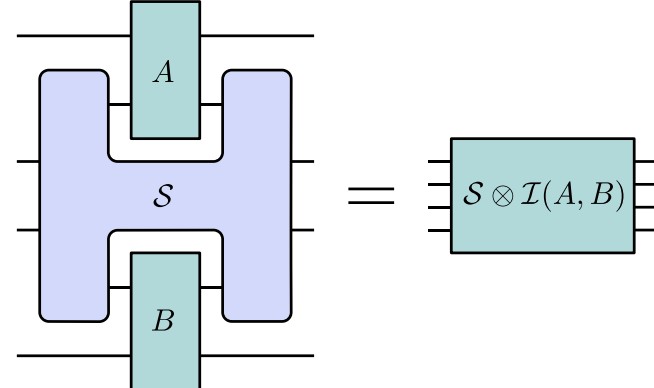

**Fig. 3 | The extended quantum switch transformation.** The quantum switch $\mathcal{S}$ is a higher-order transformation that takes as input any two quantum channels $A$ and $B$ and transforms them into another quantum channel, $\mathcal{S} \otimes \mathcal{I}(A, B)$, even when acting only on part of these channels.

simulates the quantum switch, even if the input channels are unitary. In fact, for these different orders, we prove that there does not exist any quantum circuit that can simulate the action of the quantum switch, even when acting only on unitary channels. We present more details in Supplementary Note 5.

Here, we show that an extension of the circuit in Eq. (4) allows one to perform a simulation of the quantum switch in a more general–albeit not fully general–scenario. This is the case where the quantum switch acts only on part of a quantum channel $A$ that is unitary and part of a quantum channel $B$ that is general.

In this simulation scenario, the quantum switch acts only on part of bipartite channels $A : \mathcal{L}(\mathcal{H}^{A_I} \otimes \mathcal{H}^{A'_I}) \to \mathcal{L}(\mathcal{H}^{A_O} \otimes \mathcal{H}^{A'_O})$ and $B : \mathcal{L}(\mathcal{H}^{B_I} \otimes \mathcal{H}^{B'_I}) \to \mathcal{L}(\mathcal{H}^{B_O} \otimes \mathcal{H}^{B'_O})$, both of which have two input and two output spaces. The extra input/output systems of these bipartite channels can be interpreted as environments that are not accessible to the local parties, Alice and Bob. The output channel from the quantum switch transformation is then given by $\mathcal{S} \otimes \mathcal{I}(A, B) : \mathcal{L}(\mathcal{H}^{A_I} \otimes \mathcal{H}^{c_I} \otimes \mathcal{H}^{t_I} \otimes \mathcal{H}^{B_I}) \to \mathcal{L}(\mathcal{H}^{A_O} \otimes \mathcal{H}^{c_O} \otimes \mathcal{H}^{t_O} \otimes \mathcal{H}^{B_O})$, where $\mathcal{I}$ is the identity higher-order transformation acting on the primed spaces. This transformation is depicted in Fig. 3. Since the quantum switch higher-order transformation is uniquely defined by its action on single-party unitary channels[34], there is no risk of ambiguity when considering such extended quantum switch transformations.

When the input channels $A$ and $B$ are general, i.e., not restricted to being unitary, this scenario is the strongest possible simulation scenario. In other words, a simulation $\mathcal{C}$ that is able to prepare, with some finite number of calls $k_A$ and $k_B$, a channel $\mathcal{C}(A^{\otimes k_A}, B^{\otimes k_B})$ such that

$$\mathcal{C}(A^{\otimes k_A}, B^{\otimes k_B}) = \mathcal{S} \otimes \mathcal{I}(A, B) \;\; \forall A, B, \tag{6}$$

where $A$, $B$ are arbitrary quantum channels, is a higher-order transformation that can simulate the action of the quantum switch in its most general form. We discuss further details concerning general

simulations, including also simulations of the action of the quantum switch on quantum instruments, in Supplementary Note 2.

In this general simulation scenario, we prove the following theorem, regarding the existence of a non-universal simulation of the action of the quantum switch on part of bipartite channels—in the particular case where $A$ is restricted to being a unitary channel and $B$ is a fully general quantum channel:

**Theorem 1.** The action of the quantum switch on part of bipartite quantum channels can be deterministically simulated by a quantum circuit that has access to an extra call to one the input channels, as long as that channel is restricted to being unitary.

In other words, if $A$ is a bipartite unitary channel and $B$ is a bipartite general channel, there exists a quantum circuit described by a higher-order transformation $\mathcal{C}$ that satisfies Eq. (6) for $k_A = 2$ and $k_B = 1$.

We prove this result by explicitly constructing the quantum circuit that performs this simulation, which is given by

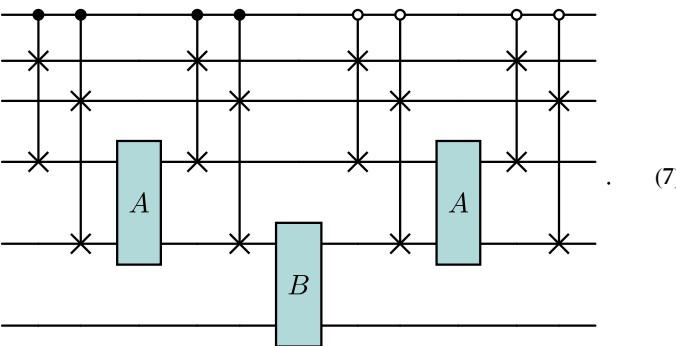

$$\tag{7}$$

Here, the first circuit line represents the qubit control, the second and third lines represent auxiliary systems, the fourth line represents Alice's primed system, the fifth line represents the target system, and the sixth line represents Bob's primed system. In Supplementary Note 2, we prove Thm. 1 explicitly, showing how it recovers the action of the quantum switch when $A$ is a unitary channel, and how it fails when $A$ is a general channel.

Given that every quantum channel can be seen as the marginal channel of a unitary channel acting on a higher-dimensional space, implemented with the use of an auxiliary system—via the Stinespring dilation theorem—it may seem counter-intuitive that the circuit in Eq. (7) fails to simulate the quantum switch for general channels. However, this is due to the fact that unitary channels acting on higher-dimensional spaces only correspond to general, non-unitary channels mapping the input to the output systems if the auxiliary system is not accessible. One way to enforce this is by requiring that the auxiliary systems be discarded. If the auxiliary system were discarded, and hence not available to be exploited during the simulation, it is straightforward to see that the circuit in Eq. (7) would fail. In Supplementary Note 2, we discuss in detail how Thm. 1 relates to the Stinespring dilation of non-unitary quantum channels.

### No-go theorems: deterministic exact simulations
We now show that the go theorem from the previous section does not generalize: not only does the circuit in Eq. (7) not work for arbitrary quantum channels $A$, but there does not exist any other quantum circuit that can simulate the quantum switch, even when a higher number of calls of the input channel $A$ are available. In order to do so, we prove an even stronger statement: one which demonstrates an exponential separation in quantum query complexity between a higher-order transformation with indefinite causal order and quantum circuits that have a fixed or classically-controlled causal order.

**Theorem 2.** There is no $(k_A + 1)$-slot higher-order transformation $\mathcal{C}$, described by a quantum circuit with fixed or classically-controlled causal order, that can simulate the quantum switch, i.e., that satisfies

$$\mathcal{C}(A^{\otimes k_A}, B) = \mathcal{S}(A, B) \tag{8}$$

for all $n$-qubit mixed unitary channels $A$ and unitary channels $B$, if

$$k_A \leq \max(2, 2^n - 1). \tag{9}$$

Therefore, such a simulation also does not exist for all $n$-qubit quantum channels $A$ and $B$.

We provide a sketch of the proof in Section "Methods", with the full proof presented in Supplementary Note 3. While Thm. 2 forbids a deterministic exact simulation of the quantum switch, it also prevents arbitrarily good probabilistic approximate simulations of it. That is because it guarantees that any probabilistic approximate simulation will necessary have a maximal probability of success $p$ strictly less than one or minimum approximation error $\epsilon$ strictly greater than zero, for any notion of metric distance.

This result can be interpreted as an exponential advantage of indefinite causal order over definite causal order for a specific task, namely that of simulating the action of the quantum switch on arbitrary quantum channels $A$ and $B$ when only one call of $B$ is available. In order to formalize this result in the context of computation, consider a classical description of a function $f$ which takes a pair of quantum channels $A$, $B$ as input and outputs a quantum channel $f(A, B)$. Consider also a higher-order transformation $\mathcal{C}$ that simulates the function $f$ deterministically and exactly—that is, such that $\mathcal{C}(A^{\otimes k_A}, B^{\otimes k_B}) = f(A, B)$ for all $A$, $B$—for some numbers $k_A$ and $k_B$ of black-box queries to the quantum channels $A$ and $B$, respectively. In the case where one of the channels is fixed to being called $k_B = 1$ times, we can define a simple notion of quantum query complexity that depends only on the number of calls to the other channel, $k_A$. We define the one-sided quantum query complexity of a function $f$, with respect to a class of higher-order quantum transformations $\mathscr{C}$, as the minimum number of queries $k_A$ while $k_B = 1$, over all higher-order quantum transformations $\mathcal{C} \in \mathscr{C}$ such that $\mathcal{C}$ simulates $f$. This definition can be seen as a step towards a fully quantum generalization of the notion of query complexity. While the standard notion of quantum query complexity has so far typically been defined for classical (e.g., boolean) functions[18], here we consider the query complexity of functions whose inputs and outputs are themselves quantum channels (see also ref. [35]). This is similar in spirit to recent works on the complexity of preparing quantum states[36,37].

In this context, we see that Thm. 2 implies that the one-sided quantum query complexity of the action of the quantum switch, with respect to all QC-CC transformations, is lower-bounded by $2^n$.

### No-go theorems: probabilistic simulations
Following from the above discussion, one might wonder whether a simulation remains impossible if more than one call of the quantum channel $B$ is allowed. Here, we also prove a no-go theorem for the simulation of the quantum switch when 2 calls of each input quantum channel is available (i.e., $k_A = k_B = 2$). In fact, we prove an even stronger result: such a simulation is impossible even for single-qubit channels, and even in a probabilistic and restricted simulation scenario.

Let us start by defining the restricted simulation scenario. Fixing the input control system to be in state $\sigma_c = |+\rangle\langle+|$ and the input target system to be in state $\rho_t = |0\rangle\langle0|$, a restricted simulation of the switch is possible if, for some finite number of calls $k_A$ and $k_B$, there exists a quantum comb or QC-CC $\mathcal{C}$ such that

$$\mathrm{tr}_{t_o}\Big(\mathcal{C}(A^{\otimes k_A}, B^{\otimes k_B})[|+\rangle\langle+| \otimes |0\rangle\langle0|]\Big)$$
$$= \mathrm{tr}_{t_o}(\mathcal{S}(A, B)[|+\rangle\langle+| \otimes |0\rangle\langle0|]) \ \forall A, B, \tag{10}$$

where $A$ and $B$ are arbitrary channels. Notice that Eq. (10) is an equality between two-qubit quantum states, namely the output control systems. Notice that, while the impossibility of a simulation in the restricted case implies the impossibility of simulation in the general case, a possibility of simulating the quantum switch in the restricted case would not imply the existence of a more general simulation.

Let us now define a probabilistic heralded simulation of the quantum switch. A probabilistic heralded simulation is a quantum comb or QC-CC $\mathcal{C}$ that, compared to the deterministic simulation in Eq. (3), outputs an extra classical bit corresponding to either the success or failure outcome of the simulation, each with a certain probability. When the value of this bit corresponds to the success outcome, the implemented simulation is exactly $\mathcal{S}(A, B)$. Such transformations can be implemented by either a quantum comb or as a QC-CC that additionally outputs a flag system encoding the success or failure outcome, followed by a dichotomic quantum measurement of the flag system. Mathematically, a probabilistic heralded simulation is a higher-order transformation $\mathcal{C} = \mathcal{C}_s + \mathcal{C}_f$, where $\mathcal{C}_s$ and $\mathcal{C}_f$ are higher-order maps that completely preserve completely positive inputs—one associated to the success and the other with the failure outcome of the transformation—and where $\mathcal{C}$ is described as either a quantum comb or as a QC-CC. In this case, $\mathcal{C} = \mathcal{C}_s + \mathcal{C}_f$ is a probabilistic simulation of the quantum switch that uses $k_A$ calls of channel $A$ and $k_B$ calls of channel $B$ if it satisfies

$$\mathcal{C}_s(A^{\otimes k_A}, B^{\otimes k_B}) = p\, \mathcal{S}(A, B) \quad \forall A, B, \quad (11)$$

where $A$ and $B$ are general quantum channels, and $p$ is the probability of a successful simulation. In the case where $\mathcal{C}$ is a quantum comb, no normalization conditions need to be imposed on $\mathcal{C}_s$. However, when $\mathcal{C}$ is a QC-CC, then normalization conditions must be imposed on $\mathcal{C}_s$ to ensure that $\mathcal{C}$ is a proper QC-CC transformation[27]. We present these constraints explicitly in Supplementary Note 1.

Combining Eqs. (10) and (11), we define a probabilistic restricted simulation of the quantum switch via

$$\mathrm{tr}_{t_o}\left(\mathcal{C}_s(A^{\otimes k_A}, B^{\otimes k_B})[|+\rangle\langle+| \otimes |0\rangle\langle0|]\right)$$
$$= p\, \mathrm{tr}_{t_o}(\mathcal{S}(A, B)[|+\rangle\langle+| \otimes |0\rangle\langle0|]) \quad \forall A, B. \quad (12)$$

It is known that any higher-order transformation with indefinite causal order can be simulated by a quantum comb in a probabilistic heralded manner with $p > 0$, even without any extra calls of the input channels[20,26]. For the particular case of the quantum switch, ref. 6 presents a probabilistic circuit based on quantum teleportation that simulates the quantum switch with $p = 1/d^2$, where $d$ is the dimension of the target system, without any extra calls. Here, we will analyze the maximal probability of success for simulating the quantum switch when extra calls are available.

As we prove in Section "Methods", for any fixed finite $d := \dim(\mathcal{H}^{t_I}) = \dim(\mathcal{H}^{t_O}) = \dim(\mathcal{H}^{A_I}) = \dim(\mathcal{H}^{A_O}) = \dim(\mathcal{H}^{B_I}) = \dim(\mathcal{H}^{B_O})$, $k_A$, and $k_B$, the maximum probability of success $p$ of simulating the quantum switch acting on all general quantum channels $A$ and $B$ can be computed with a semidefinite program (SDP). This is because if a simulation exists for a finite subset of channels that form a basis for the linear subspace spanned by $k_A$ copies of a quantum channel $A$, and equivalently for $B$, then said simulation is also valid for all channels $A$ and $B$, due to the linearity of higher-order transformations. Crucially, in the case where $\mathcal{C}$ is a quantum comb, the maximum probability of success of a simulation of the quantum switch may depend on the order of the input channels $A$ and $B$. In the case where $\mathcal{C}$ is a QC-CC, all possible fixed and dynamical orders are automatically optimized over.

We are now ready to state this result (Table 1).

**Theorem 3.** There is no quantum circuit that can deterministically simulate the action of the quantum switch on all quantum channels

**Table 1 | Maximum probability of a restricted simulation of the qubit quantum switch**

| $(k_A, k_B)$ | order | probability |
|---|---|---|
| (1, 1) | AB | $p < \frac{4001}{10000}$ |
| | AAB | $p < \frac{5715}{10000}$ |
| (2, 1) | ABA | $p < \frac{4919}{10000}$ |
| | BAA | $p < \frac{5001}{10000}$ |
| | AABB | $p < \frac{8307}{10000}$ |
| (2, 2) | ABAB | $p < \frac{8484}{10000}$ |
| | ABBA | $p < \frac{8695}{10000}$ |
| | AAAB | $p < \frac{8373}{10000}$ |
| (3, 1) | AABA | $p < \frac{6909}{10000}$ |
| | ABAA | $p < \frac{7597}{10000}$ |
| | BAAA | $p < \frac{6845}{10000}$ |

Rigorous upper bounds for the maximum probability $p$ of a restricted simulation of the quantum switch acting on a pair of arbitrary qubit quantum channels $A$, $B$, using open-slot quantum circuits that take $k_{A(B)}$ calls of an arbitrary quantum channel $A(B)$ in a certain order. All bounds were derived via computer-assisted proofs.

when $(k_A, k_B) \in \{(1, 1), (2, 1), (2, 2), (3, 1)\}$, where $k_A$ is the number of calls of input channel $A$ and $k_B$ is the number of calls of input channel $B$.

Moreover, even for a restricted simulation—i.e., for fixed input systems and discarded output target system—the action on the quantum switch on general qubit channels, when $(k_A, k_B) \in \{(1, 1), (2, 1), (2, 2), (3, 1)\}$, can be simulated with at most a probability $p < 1$, with upper bounds given in Table 1.

We prove this theorem using a method that we develop for computer-assisted proofs that transforms the numerically imperfect solution of an SDP into a rigorous upper bound for the probability of success that can be expressed in terms of rational numbers. The method is presented in Section "Methods" with further details provided in Supplementary Note 6.

Still in the restricted simulation scenario, we found three different particular cases where a deterministic simulation is possible for qubit channels, but nevertheless impossible in the more general scenario. In the following, we discuss these three cases, with more details being provided in Supplementary Note 4.

The first case of this kind is when the quantum switch acts on arbitrary, yet identical, qubit quantum channels, i.e., $A = B$. Although the input channels in this case are identical, it is straightforward to see from Eq. (1) that the action of the switch is non-trivial, namely, that the output channel $\mathcal{S}(A, A)$ is not equivalent to $A$ applied twice on the target system when $A$ is not a unitary channel. A remarkable example of this point is the case where $A$ is the depolarizing channel[38]. We find that a restricted deterministic quantum comb simulation exists for qubit channels in the "AAAA" scenario, i.e., with 4 identical calls of the arbitrary qubit quantum channel $A$. Since in this case the maximum probability of success is $p = 1$, one cannot certify this result with a computer-assisted proof, which can yield upper and lower bounds for $p$ with arbitrary yet only finite precision. Instead, we obtain this result by numerically evaluating an SDP with very high precision, ensuring that all positivity constraints are strictly satisfied and that all equality constraints are satisfied up to an error of at most $10^{-9}$ in the operator norm. Furthermore, 4 identical calls of the arbitrary qubit quantum channel $A$ are not only sufficient, but necessary for a deterministic restricted simulation: if only 2 or 3 calls are allowed, we prove upper bounds for the maximum probability of simulation that are always strictly less than 1 (see Table 2). However, this only holds in the restricted simulation case. When the output target system is not discarded, we find numerically that a deterministic simulation of the AAAA scenario is no longer possible (see

**Table 2 | Bounds for the maximum probability of a restricted simulation of the qubit quantum switch acting on identical channels**

| k | order | probability |
|---|-------|-------------|
| 2 | AA | $p < \frac{4001}{10000}$ |
| 3 | AAA | $p < \frac{6534}{10000}$ |
| 4 | AAAA | $p = 1$ (*) |

Rigorous upper bounds for the maximum probability $p$ of a restricted simulation of the quantum switch that acts on a pair of identical qubit quantum channels, using open-slot quantum circuits that take $k \in \{2, 3\}$ calls of an arbitrary quantum channel $A$. All bounds were derived via computer-assisted proofs. (*) For the case of $k = 4$, we numerically find that a deterministic simulation is possible, i.e., that $p = 1$ up to a numerical precision of $10^{-9}$.

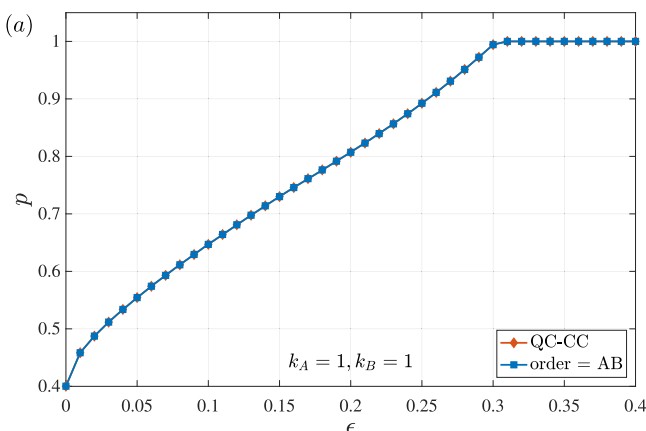

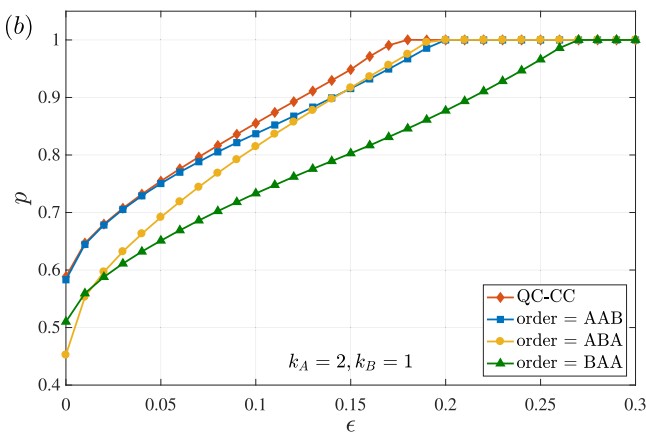

**Fig. 4 | Robustness of the impossibility of simulation.** Maximum probability of success $p$ of simulating any higher-order transformation that is $\epsilon$-close to the quantum switch using quantum combs—of order AAB, ABA, or BAA—or quantum circuits with classical control (QC-CC), as a function of $\epsilon$. In plot (**a**), we show the case where $k_A = k_B = 1$. Note how the QC-CC simulation curve numerically coincides with the comb simulation curve. In plot (**b**), we show the case where $k_A = 2$ and $k_B = 1$. In this case, for some values of $\epsilon$, QC-CCs show an advantage with respect to quantum combs.

Supplementary Note 4 for more details). This shows how highly nontrivial the action of the quantum switch is even when acting on identical quantum channels.

The other two restricted cases where a deterministic restricted simulation is possible concern qubit unitary channels applied in the order AABB and BAAA, two cases where trivial extensions of the circuit in Eq. (4) fail. Similarly to the case of identical qubit channels, such simulations no longer exist if the output target system is not discarded. More details in Supplementary Note 4.

## No-go theorems: approximate simulations
By exhibiting concrete upper bounds for the probability of successful simulation of the quantum switch that are significantly below 1, we demonstrate the extent of the robustness of our results with respect to a figure of merit related to a probabilistic simulation. Another pertinent notion of robustness in this context is that of an approximate simulation—although the quantum switch may not be able to be exactly simulated, it could still be the case that other higher-order transformations that are $\epsilon$-close to the quantum switch can be simulated. This question is particularly relevant for experimental scenarios, where noise and imprecision hinder the ability of preparing any specific transformation exactly. Here, we prove that even in the approximate case, a simulation of the quantum switch is not possible for some particular number of calls $(k_A, k_B)$. We do so by exhibiting values of $\epsilon$ that are significantly above zero, for which even a probabilistic simulation, for some $p < 1$, that we also exhibit, is not possible.

For an approximate simulation of the quantum switch, it is useful to consider a partly restricted scenario that is required to hold for fixed input control and target systems, but without discarding the output target system. Similarly to the restricted simulation case, proving the impossibility of a simulation in this partly restricted case also implies the impossibility of a simulation in more general scenarios. Let $\sigma_c = |+\rangle\langle+|$ be the state of the input control system, $\rho_t = |0\rangle\langle0|$ be the state of the input target system, and $\widetilde{S}$ be a higher-order transformation that acts on the same space as the quantum switch. We define a simulation $\mathcal{C}$ to be $\epsilon$-close to the quantum switch $\mathcal{S}$ in this scenario if there exists an $\widetilde{S}$ such that

$$
\begin{aligned}
&\mathcal{C}(A^{\otimes k_A}, B^{\otimes k_B})[|+\rangle\langle+| \otimes |0\rangle\langle0|]\\
&= \widetilde{\mathcal{S}}(A,B)[|+\rangle\langle+| \otimes |0\rangle\langle0|] \quad \forall A, B,
\end{aligned}
\tag{13}
$$

where $A$ and $B$ are arbitrary quantum channels and, for $\mathcal{S}_{+0}(\cdot, \cdot) := \mathcal{S}(\cdot, \cdot)[|+\rangle\langle+| \otimes |0\rangle\langle0|]$ and $\widetilde{\mathcal{S}}_{+0}(\cdot, \cdot) := \widetilde{\mathcal{S}}(\cdot, \cdot)[|+\rangle\langle+| \otimes |0\rangle\langle0|]$, it holds that

$$
F(\mathcal{S}_{+0}, \widetilde{\mathcal{S}}_{+0}) \geq 1 - \epsilon,
\tag{14}
$$

where $F(\mathcal{M}, \mathcal{N})$ is the normalized fidelity between the Choi operators associated to the higher-order transformations $\mathcal{M}$ and $\mathcal{N}$ (see Supplementary Note 1).

Combining probabilistic heralded and approximate protocols of simulation, for a fixed $\epsilon$, $(k_A, k_B)$, and $d$, the maximum probability of successful simulation can also be computed via an SDP. We show that, for a range of values of $\epsilon > 0$, the probability of simulating the quantum switch in this partly restricted scenario is $p < 1$ for the cases where $(k_A, k_B) \in \{(1, 1), (2, 1)\}$, considering simulations using quantum combs of all possible orders as well as QC-CCs. We present our numerical findings in Fig. 4 for the case where $d = 2$. In the first plot (Fig. 4a), where $(k_A, k_B) = (1, 1)$, the QC-CC simulation curve numerically coincides with the quantum comb simulation curve. Here, $p < 1$ for $0 \leq \epsilon \lesssim 0.3$. In the second plot (Fig. 4b), where $(k_A, k_B) = (2, 1)$, each of the three possible quantum comb orders and QC-CC yield different values for the maximum probability of success for different values of $\epsilon$. Here, $p < 1$ for $0 \leq \epsilon \lesssim 0.18$, after which point a QC-CC simulation exists. Notice that for an exact simulation, i.e., when $\epsilon = 0$, a QC-CC simulation yields a probability of success that coincides with the highest among all possible orders, but as $\epsilon$ increases, it shows an advantage in the probability of success as compared to quantum combs.

## A conjecture
Taken together, our results indicate the hardness of simulating the quantum switch. Considering all the evidence we have gathered here supporting a high cost, if not altogether the impossibility, of simulating the quantum switch deterministically, we are motivated to propose the following conjecture:

**Conjecture 1.** There is no $(k_A + k_B)$-slot higher-order transformation $\mathcal{C}$, described by a quantum circuit with fixed or classically-controlled causal order, that can simulate the quantum switch, i.e., that satisfies

$$\mathcal{C}(A^{\otimes k_A}, B^{\otimes k_B}) := \mathcal{S}(A, B) \qquad (15)$$

for all $n$-qubit quantum channels $A$ and $B$, if $\max(k_A, k_B) \leq g(n)$, for some $g(n) = \Theta(2^n)$.

Our rationale behind conjecturing that no simulation is possible, even with multiple (albeit a sub-exponential number of) calls to both input channels is the following. As mentioned above, there exists a deterministic and exact simulation of the quantum switch with a single query to a general channel $B$ and two queries to a unitary channel $A$. However, simulating the quantum switch for general channels $A$ and $B$ requires correlating each Kraus operator $A_k$ on the $|0\rangle$ branch—which can be obtained by querying $A$ before $B$—with the same $A_k$ on the $|1\rangle$ branch—which can be obtained by querying $A$ after $B$ [see Eq. (2)]. In this view, $B$ can be considered as a fixed channel[39], and therefore the intuition is that querying it multiple times is no better than querying it once. The rational for the bound to be $\Theta(2^n)$ is that all the main steps in the proof of Thm. 2 except one hold for a bound of $\Theta(2^n)$ with $(k_A + k_B)$-slot higher-order transformations, and only for Lemma 4 were we only able to prove the $(k_A + 1)$-slot case. It remains to be seen whether the action of the quantum switch can be simulated with any finite number of queries to one or both input channels.

## Discussion

In this work, we have concretely formulated the problem of simulating higher-order transformations with indefinite causal order using higher-order transformations that obey causal constraints and have access to more calls of the input operations. Our case study for this problem was the quantum switch. We defined the problem in its full generality, considering also cases where the quantum switch acts only on part of its input quantum channels.

We showed that the (one-sided) quantum query complexity of the action of the quantum switch on $n$-qubit channels, with respect to all quantum circuits with fixed and classically-controlled causal order, is lower bounded by $2^n$ (Thm. 2). Notably, the exponential separation that we prove is formulated with respect to a computational task where the inputs and outputs of the computation are given by black-box quantum channels[1,2,6,40]. This is in contrast to previous works on the query complexity of the quantum switch, where the output of the computation is a bit representing the evaluation of a classical function, in which case no such exponential separation has been found[9,16–18,41].

We additionally proved that an extra copy of each input quantum channel, i.e., $(k_A, k_B) = (2, 2)$, is not sufficient to deterministically simulate the quantum switch with a quantum circuit. Considering probabilistic simulations, we proved a range of rigorous strict upper bounds on the probability of successful simulation in various different scenarios (Thm. 3). In order to do so, we developed and applied a general method of computer-assisted proofs based on semidefinite programming. We also explored approximate simulations in some cases, showing that the switch cannot be simulated even approximately for some $\epsilon > 0$ with probability $p = 1$.

Finally, we showed that for some particular kinds of input, non-universal simulations exist. The most general such case is when $A$ is a bipartite unitary channel, $B$ is a bipartite general channel, and the quantum switch acts only on part of the input channels. Such a simulation exists and requires only an extra call of $A$ (Thm. 1). A simulation also exists for the particular case where the input pair of channels of the switch are identical qubit quantum channels, $A = B$. Here, 4 calls of the input channel are necessary and sufficient for a restricted deterministic simulation, where the input states are fixed and the target output system is discarded. However, this result does not hold for more general simulation scenarios, where the output target system is not discarded. This possibility of simulation in the restricted case, but not in more general cases, is also true for the simulation of the action of the switch on qubit unitary channels with the orders AABB and BAAA.

Our results have implications for the analysis of the experiments based on the quantum switch[42], particularly the ones that make use of non-unitary channels, such as those reported in refs. 43–47. A long-standing debate in the community is whether these experiments are a genuine realization or some form of simulation of the quantum switch. One important criticism is that several experimental setups leave room for the interpretation that there are two independent calls of each input quantum channel available in the transformation. Here, we have proven that such access still does not allow for a causally ordered explanation of the observed data. On the other hand, an alternative causal model that can reproduce the action of the quantum switch is one that, instead of considering the inputs to be two independent calls of a general channel, considers inputs that can be described as bipartite channels with memory, which can be interpreted as two correlated uses of a general quantum channel[48,49]. As a corollary of our result in Theorem 3, we observe that there does not exist a quantum circuit that can take two independent uses of an arbitrary channel as input and output a bipartite channel with memory that corresponds to two correlated uses of the input quantum channels. Therefore, a causal model based on correlated inputs is not compatible with a simulation scenario that considers black-box inputs, as considered here. However, this point alone is not sufficient to rule out this model as a fair description of the experiments. It is our hope that our results will encourage further rigorous analysis of potential loopholes in the quantum switch experiments.

Our work opens up the study of query complexity in the context of higher-order quantum computation where the inputs and outputs of the computation are general quantum channels. The question of whether any finite number of calls suffices to perform a simulation of the switch is of high relevance and remains open. Further investigation of this topic will be crucial to determine, for example, whether advantages in specific tasks that have been previously demonstrated can persist in the asymptotic limit of the available number of calls. Should it ever be shown that the quantum switch can be simulated with a finite number of calls, a relevant follow-up question is related to the scaling of the necessary number of calls, and to whether or not the switch can be efficiently simulated in a deterministic setting.

The existence of efficient deterministic and exact simulation of quantum processes with indefinite causal order may pose itself as an interesting physical guideline to help us understand on the one hand, which kinds of higher-order transformations can be realistically implemented, and on the other hand, what advantages they may provide.

## Methods
### Preliminaries
In our technical calculations, we make use of the Choi–Jamiołkowski isomorphism[50,51] and the link product[1].

Using this isomorphism, any linear map $M : \mathcal{L}(\mathcal{H}^I) \to \mathcal{L}(\mathcal{H}^O)$ that maps linear operators acting on an input space to linear operators acting on an output space can be represented by a linear operator $J^M \in \mathcal{L}(\mathcal{H}^I \otimes \mathcal{H}^O)$ that acts on the joint input and output space, called the Choi operator, which is given by

$$J^M := \sum_{ij} |i\rangle \langle j| \otimes M[|i\rangle \langle j|], \qquad (16)$$

where $\{|i\rangle\}_i$ is the computational basis. A linear map $M$ is completely positive (CP) if and only if its Choi operator $J^M$ is positive semidefinite,

and is trace-preserving (TP) if and only if $J^M$ satisfies $\mathrm{tr}_O(J^M) = \mathbb{1}^I$, where $\mathbb{1}^I$ is the identity operator in $\mathcal{L}(\mathcal{H}^I)$. Hence, the Choi operator of general quantum channels satisfy these conditions.

Any linear operator $\mathsf{U}: \mathcal{H}^I \to \mathcal{H}^O$ can be represented by its Choi vector $|\mathsf{U}\rangle\rangle \in \mathcal{H}^I \otimes \mathcal{H}^O$, where

$$|\mathsf{U}\rangle\rangle := \sum_i |i\rangle \otimes \mathsf{U}|i\rangle. \tag{17}$$

Hence, a unitary channel $U$ such that $U[\rho] = \mathsf{U}\rho\mathsf{U}^\dagger$, where $\mathsf{U}$ is a unitary operator that corresponds to the single Kraus operator of the channel $U$, has Choi operator $J^U = |\mathsf{U}\rangle\rangle\langle\langle\mathsf{U}|$.

Using this representation, the composition $G \circ F$ of two maps $F: \mathcal{L}(\mathcal{H}^1) \to \mathcal{L}(\mathcal{H}^2)$ and $G: \mathcal{L}(\mathcal{H}^2) \to \mathcal{L}(\mathcal{H}^3)$, with respective Choi operators $J^F \in \mathcal{L}(\mathcal{H}^1 \otimes \mathcal{H}^2)$ and $J^G \in \mathcal{L}(\mathcal{H}^2 \otimes \mathcal{H}^3)$, is given by the link product $J^F * J^G \in \mathcal{L}(\mathcal{H}^1 \otimes \mathcal{H}^3)$, according to

$$J^F * J^G := \mathrm{tr}_2[(J^F \otimes \mathbb{1}^3)(\mathbb{1}^1 \otimes (J^G)^{T_2})], \tag{18}$$

where $(\cdot)^{T_X}$ denotes partial transposition on the space $\mathcal{L}(\mathcal{H}^X)$.

The link product for Choi vectors $|\mathsf{Q}\rangle\rangle \in \mathcal{H}^1 \otimes \mathcal{H}^2$ and $|\mathsf{R}\rangle\rangle \in \mathcal{H}^2 \otimes \mathcal{H}^3$ is given by $|\mathsf{Q}\rangle\rangle*|\mathsf{R}\rangle\rangle \in \mathcal{H}^1 \otimes \mathcal{H}^3$, defined as

$$|\mathsf{Q}\rangle\rangle*|\mathsf{R}\rangle\rangle := \sum_i (\mathbb{1}^1 \otimes \langle i|)|\mathsf{Q}\rangle\rangle \otimes (\langle i| \otimes \mathbb{1}^3)|\mathsf{R}\rangle\rangle. \tag{19}$$

## Sketch of the proof of Theorem 2

While we provide the full proof of Thm. 2 in Supplementary Note 3, here we give a sketch of the proof for the case where $k_A = 2$.

Here we adopt a slightly different notation from the main text, and write the higher-order transformation $\mathcal{C}$ as a function $\mathcal{C}(A, B, C)$ that takes three arguments, each corresponding to a quantum channel that can be plugged into one of its three slots. Hence, instead of writing, e.g., $\mathcal{C}(A^{\otimes 2}, B)$ we will use $\mathcal{C}(A, A, B)$.

Let $\mathcal{C}(A, A, B)$ be a simulation of the action of the quantum switch $\mathcal{S}(A, B)$ on all mixed unitary quantum channels $A$ and unitary channels $B$. Then, for arbitrary unitary channels $U_1$, $U_2$, $V$, the simulation $\mathcal{C}$ necessarily respects

$$\mathcal{C}\left(\frac{U_1+U_2}{2}, \frac{U_1+U_2}{2}, V\right) = \mathcal{S}\left(\frac{U_1+U_2}{2}, V\right). \tag{20}$$

By linearity, Eq. (20) implies that

$$\sum_{i,j=1}^2 \mathcal{C}(U_i, U_j, V) = 2\sum_{k=1}^2 \mathcal{S}(U_k, V). \tag{21}$$

Let $C$ be the Choi operator of the simulation $\mathcal{C}$, $S = |\mathsf{S}\rangle\rangle\langle\langle\mathsf{S}|$ be the Choi operator of the quantum switch transformation $\mathcal{S}$, defined by Eq. (1) and written explicitly in Supplementary Note 1, and $|\mathsf{U}_l\rangle\rangle\langle\langle\mathsf{U}_l|$ denote the Choi operator of a unitary channel $U_l$. Then, Eq. (21) can be expressed as

$$\sum_{i,j=1}^2 C*\left(|\mathsf{U}_i\rangle\rangle\langle\langle\mathsf{U}_i| \otimes |\mathsf{U}_j\rangle\rangle\langle\langle\mathsf{U}_j| \otimes |\mathsf{V}\rangle\rangle\langle\langle\mathsf{V}|\right) $$
$$= 2\sum_{k=1}^2 |\mathsf{S}\rangle\rangle\langle\langle\mathsf{S}|*(|\mathsf{U}_k\rangle\rangle\langle\langle\mathsf{U}_k| \otimes |\mathsf{V}\rangle\rangle\langle\langle\mathsf{V}|). \tag{22}$$

Assuming that $C$ is positive semidefinite, it accepts a decomposition $C = \sum_a |C^{(a)}\rangle\rangle\langle\langle C^{(a)}|$ where $|C^{(a)}\rangle\rangle\langle\langle C^{(a)}| \geq 0$ for all $a$. From the

positivity of $|\mathsf{U}_l\rangle\rangle\langle\langle\mathsf{U}_l|$ and $|\mathsf{V}\rangle\rangle\langle\langle\mathsf{V}|$, it follows that

$$\left|C^{(a)}\right\rangle\rangle\langle\langle C^{(a)}|^*(|\mathsf{U}_1\rangle\rangle\langle\langle\mathsf{U}_1| \otimes |\mathsf{U}_2\rangle\rangle\langle\langle\mathsf{U}_2| \otimes |\mathsf{V}\rangle\rangle\langle\langle\mathsf{V}|)$$
$$\leq 2|\mathsf{S}\rangle\rangle\langle\langle\mathsf{S}|^*[(|\mathsf{U}_1\rangle\rangle\langle\langle\mathsf{U}_1| + |\mathsf{U}_2\rangle\rangle\langle\langle\mathsf{U}_2|) \otimes |\mathsf{V}\rangle\rangle\langle\langle\mathsf{V}|], \tag{23}$$

for every $a$.

Following the definition of the link product for Choi vectors, $|\mathsf{Q}\rangle\rangle\langle\langle\mathsf{Q}|^*|\mathsf{R}\rangle\rangle\langle\langle\mathsf{R}|$ is given by $(|\mathsf{Q}\rangle\rangle^*|\mathsf{R}\rangle\rangle)(|\mathsf{Q}\rangle\rangle^*|\mathsf{R}\rangle\rangle)^\dagger$ (see, e.g., Lemma 1 of ref. 52). Thus, the support of the right-hand side of Eq. (23) is given by $\mathrm{span}\{|\mathsf{S}\rangle\rangle^*(|\mathsf{U}_k\rangle\rangle \otimes |\mathsf{V}\rangle\rangle)\}_{k=1}^2$ and that of the left-hand side of Eq. (23) is the one-dimensional subspace spanned by $\left|C^{(a)}\right\rangle\rangle^*(|\mathsf{U}_1\rangle\rangle \otimes |\mathsf{U}_2\rangle\rangle \otimes |\mathsf{V}\rangle\rangle)$. Therefore, one can write

$$\left|C^{(a)}\right\rangle\rangle^*(|\mathsf{U}_1\rangle\rangle \otimes |\mathsf{U}_2\rangle\rangle \otimes |\mathsf{V}\rangle\rangle)$$
$$= \sum_{k=1}^2 \xi_k^{(a)}(\mathsf{U}_1, \mathsf{U}_2, \mathsf{V})|\mathsf{S}\rangle\rangle^*(|\mathsf{U}_k\rangle\rangle \otimes |\mathsf{V}\rangle\rangle), \tag{24}$$

for some coefficients $\xi_k^{(a)}(\mathsf{U}_1, \mathsf{U}_2, \mathsf{V}) \in \mathbb{C}$. A proof of this fact is in Lemmas 1 and 2 in Supplementary Note 3.

We now invoke Lemma 3 and Lemma 4 in Supplementary Note 3 to ensure that, when Eq. (24) is satisfied, there exist vectors $|\xi_1^{(a)}\rangle\rangle \in \mathcal{H}^{I_2} \otimes \mathcal{H}^{O_2}$ and $|\xi_2^{(a)}\rangle\rangle \in \mathcal{H}^{I_1} \otimes \mathcal{H}^{O_1}$, such that

$$\left|C^{(a)}\right\rangle\rangle = \sum_{k=1}^2 |\mathsf{S}\rangle\rangle \otimes |\xi_k^{(a)}\rangle\rangle, \tag{25}$$

for all $a$, where $|\xi_1^{(a)}\rangle\rangle$ and $|\xi_2^{(a)}\rangle\rangle$ are independent of $\mathsf{U}_1$, $\mathsf{U}_2$, and $\mathsf{V}$.

Next, we argue why this is the case.

The basic idea for this part of the proof is based on differentiation with respect to a parametrization of the input unitary operators, a technique introduced in ref. 35 by some of the present authors. Suppose that $\mathsf{U}_1$, $\mathsf{U}_2$, and $\mathsf{V}$ are taken from the set $\{I, X, Y, Z\}$ of Pauli operators. If $\mathsf{U}_1 \neq \mathsf{U}_2$, then $|\mathsf{S}\rangle\rangle^*(|\mathsf{U}_1\rangle\rangle \otimes |\mathsf{V}\rangle\rangle)$ and $|\mathsf{S}\rangle\rangle^*(|\mathsf{U}_2\rangle\rangle \otimes |\mathsf{V}\rangle\rangle)$ are linearly independent. In this case, we can show using linearity that $\xi_1^{(a)}(\mathsf{U}_1, \mathsf{U}_2, \mathsf{V})$ is independent of $\mathsf{U}_1$, $\mathsf{V}$ and $\xi_2^{(a)}(\mathsf{U}_1, \mathsf{U}_2, \mathsf{V})$ is independent of $\mathsf{U}_2$, $\mathsf{V}$. If on the other hand $\mathsf{U}_1 = \mathsf{U}_2 = \sigma$, then $|\mathsf{S}\rangle\rangle^*(|\mathsf{U}_1\rangle\rangle \otimes |\mathsf{V}\rangle\rangle)$ and $|\mathsf{S}\rangle\rangle^*(|\mathsf{U}_2\rangle\rangle \otimes |\mathsf{V}\rangle\rangle)$ are not linearly independent.

In such cases, it turns out that $\xi_1^{(a)}(\sigma, \sigma, \mathsf{V})$ and $\xi_2^{(a)}(\sigma, \sigma, \mathsf{V})$ can be suitably chosen as $\xi_1^{(a)}(\sigma', \sigma, \mathsf{V})$ and $\xi_2^{(a)}(\sigma, \sigma', \mathsf{V})$, respectively, where $\sigma' \neq \sigma$ is a Pauli operator. Note that $\xi_1^{(a)}(\sigma', \sigma, \mathsf{V})$ and $\xi_2^{(a)}(\sigma, \sigma', \mathsf{V})$ do not depend on the choice of $\sigma'$ as long as $\sigma' \neq \sigma$ holds. The fact that such a redefinition is consistent with Eq. (24) can be proven by differentiating the expression $\xi_i^{(a)}(\tilde{\sigma}(\theta), \tilde{\sigma}(\theta), \mathsf{V})$, where $\tilde{\sigma}(\theta)$ is a parameterized unitary operator satisfying $\tilde{\sigma}(0) = \sigma$ and $\frac{d}{d\theta}|_{\theta=0}\tilde{\sigma}(\theta) \propto \sigma'$. This redefinition implies that $\xi_1^{(a)}(\mathsf{U}_1, \mathsf{U}_2, \mathsf{V})$ and $\xi_2^{(a)}(\mathsf{U}_1, \mathsf{U}_2, \mathsf{V})$ are independent of $\mathsf{U}_1$ and $\mathsf{U}_2$, respectively. By linearity, we can show that for $i \in \{1, 2\}$, $\xi_k^{(a)}(\mathsf{U}_1, \mathsf{U}_2, \mathsf{V})$ is independent of $\mathsf{V}$.

The independence relations above imply that we can write $\xi_1^{(a)}(\mathsf{U}_1, \mathsf{U}_2, \mathsf{V}) = |\xi_1^{(a)}\rangle\rangle^*|\mathsf{U}_2\rangle\rangle$ and $\xi_2^{(a)}(\mathsf{U}_1, \mathsf{U}_2, \mathsf{V}) = |\xi_2^{(a)}\rangle\rangle^*|\mathsf{U}_1\rangle\rangle$ for some vectors $|\xi_1^{(a)}\rangle\rangle$, $|\xi_2^{(a)}\rangle\rangle$. Substituting this into Eq. (24) gives

$$\left|C^{(a)}\right\rangle\rangle^*(|\mathsf{U}_1\rangle\rangle \otimes |\mathsf{U}_2\rangle\rangle \otimes |\mathsf{V}\rangle\rangle)$$
$$= \sum_{k=1}^2 |\mathsf{S}\rangle\rangle \otimes |\xi_k^{(a)}\rangle\rangle^*(|\mathsf{U}_1\rangle\rangle \otimes |\mathsf{U}_2\rangle\rangle \otimes |\mathsf{V}\rangle\rangle). \tag{26}$$

Since this holds for all combinations of Pauli operators $\mathsf{U}_1, \mathsf{U}_2, \mathsf{V}$, we obtain Eq. (25).

Hence, we have shown that the assumption of a simulation of the quantum switch given by a higher-order transformation with Choi operator $C = \sum_a |C^{(a)}\rangle\rangle\langle\langle C^{(a)}| \geq 0$ implies the existence of vectors $|\xi_1^{(a)}\rangle\rangle$ and $|\xi_2^{(a)}\rangle\rangle$ that satisfy Eq. (25). Finally, we invoke Lemma 5 in

Supplementary Note 3, which states that such a higher-order transformation does not satisfy the QC-CC conditions.

## Semidefinite programming

Here, we show that the problem of simulating the quantum switch can be phrased as an SDP.

Let $S$ be the Choi operator of the quantum switch transformation $\mathcal{S}$, defined in Eq. (1), written explicitly in the Supplementary Note 1. Let $J^A$ and $J^B$ be the Choi operators of the quantum channels $A$ and $B$. Finally, let $C$ be the Choi operator of the deterministic higher-order transformation $\mathcal{C}$, which corresponds to a quantum comb, and $C_s$ be the Choi operator of the higher-order transformation $\mathcal{C}_s$ associated with the success outcome of a probabilistic transformation. Then, the maximum probability of simulating the action of the quantum switch $S$ on a set of $N_A$ channels $\{J_i^A\}_{i=1}^{N_A}$ and a set of $N_B$ channels $\{J_j^B\}_{j=1}^{N_B}$ using a quantum comb $C$ that acts on $k_A$ calls of $J_i^A$ and $k_B$ calls of $J_j^B$ is given by the following SDP:

$$
\begin{aligned}
&\textbf{given} \quad \{J_i^A\}_i, \{J_j^B\}_j, k_A, k_B \\
&\textbf{max} \qquad\qquad p \\
&\textbf{s.t.} \quad C_s * [(J_i^A)^{\otimes k_A} \otimes (J_j^B)^{\otimes k_B}] = p\,S^*(J_i^A \otimes J_j^B) \ \forall i,j \\
&\qquad\quad C_s \geq 0, \ C - C_s \geq 0, \\
&\qquad\quad \mathbb{P}(C) = C, \ \text{tr}(C) = d_{c_i} d_{t_i} d_{A_O}^{k_A} d_{B_O}^{k_B},
\end{aligned}
\tag{27}
$$

where $\mathbb{P}$ is the projector onto the linear subspace spanned by valid quantum combs[1,5,8] (explicitly written in Supplementary Note 1), and $d_X := \dim(\mathcal{H}^X)$. For a simulation given by QC-CC transformations, the constraint $\mathbb{P}(C) = C$ should be substituted by the appropriate linear constraints that define a proper QC-CC transformations, as well as additional normalization constraints on $C_s$[27]. These constraints are explicitly written in the Supplementary Note 1 for the cases of 2 and 3 slots. Notice that, following results from Section "Discussion" of ref. 20, the variables $C$ and $C_s$ in this SDP can be restricted to the field of real numbers without loss of generality, a feature that improves numerical performance.

The dual problem associated to this SDP can be found by combining standard methods[53] with dual affine techniques (see, e.g., the derivation of a similar dual problem in Appendix B of ref. 10, inspired by ref. 54). It is given by

$$
\begin{aligned}
&\textbf{given} \quad \{J_i^A\}_i, \{J_j^B\}_j, k_A, k_B \\
&\textbf{min} \quad \frac{1}{d_{A_I}^{k_A} d_{B_I}^{k_B} d_{c_O} d_{t_O}} \text{tr}(\Gamma) \\
&\textbf{s.t.} \quad \sum_{i,j} \text{tr}[R_{ij}(S^*(J_i^A \otimes J_j^B))] = 1 \\
&\qquad\quad \Gamma - \sum_{i,j} R_{ij} \otimes [(J_i^A)^{\otimes k_A} \otimes (J_j^B)^{\otimes k_B}]^T \geq 0 \\
&\qquad\quad \Gamma \geq 0, \ \overline{\mathbb{P}}(\Gamma) = \Gamma,
\end{aligned}
\tag{28}
$$

where $\overline{\mathbb{P}}$ is the projector onto the linear subspace spanned by the dual affine of valid combs[5,10,54] (also explicitly written in the Supplementary Note 1). In the case of this dual SDP problem, the variable $\Gamma$ may also be restricted to the field of real numbers, since it is the Lagrange multiplier associated to the constraint $\mathbb{P}(C) = C$ (an equality between real matrices); however, the same is not true for the variables $\{R_{ij}\}$, which are the Lagrange multipliers associated to the constraints $C_s * [(J_i^A)^{\otimes k_A} \otimes (J_j^B)^{\otimes k_B}] = p\,S^*(J_i^A \otimes J_j^B)$, an equality between complex matrices. Nonetheless, restricting $\{R_{ij}\}$ to real values still allows for feasible points that yield a valid upper bound for the optimal solution of this problem.

Finally, the SDPs 27 and (28) satisfy the condition of strong duality, which is implied by the fact that a strictly feasible point for the primal problem can be created from the probabilistic simulation of ref. 6 that requires no extra calls.

## Basis design

For any input set of channels, the solution of SDP 27, or equivalently of SDP (28), corresponds to the maximum probability of success of a simulation of the action of the quantum switch on the specific given inputs, with a simulation that has access to the specified number of calls of the input channels and that satisfies the specified causal constraints. Then, the maximum probability of success of a universal switch simulation—one that works for all possible input channels and not only for the given inputs—can be obtained by setting $\{(J_i^A)\}_i$ to be a set of operators that forms a basis for the span $(\{(J_i^A)^{\otimes k_A}\}_i)$, i.e., the subspace spanned by $k_A$ copies of an arbitrary quantum channel, and equivalently for $\{(J_j^B)\}$. In this case, if the constraint $C_s^*[(J_i^A)^{\otimes k_A} \otimes (J_j^B)^{\otimes k_B}] = p\,S^*(J_i^A \otimes J_j^B)$ holds for all $i,j$, then it also holds for arbitrary quantum channels. In other words, it implies that $C_s^*[(J^A)^{\otimes k_A} \otimes (J^B)^{\otimes k_B}] = p\,S^*(J^A \otimes J^B)$ holds for all channels $J^A$ and $J^B$, thereby implying the existence of a universal probabilistic simulator with success probability $p$.

There are a few properties that the elements $J_i^A$ and $J_j^B$ of these bases must satisfy in order to be valid inputs of the SDPs 27 and (28). The first is that it is necessary to ensure that the operators $J_i^A$ and $J_j^B$ individually correspond to TP maps, in order for the total trace of both sides of the constraint $C_s * [(J_i^A)^{\otimes k_A} \otimes (J_i^B)^{\otimes k_B}] = p\,S * (J_i^A \otimes J_j^B)$ to match. However, they do not need to correspond to CP maps, i.e., to be positive semidefinite operators. That is because all elements of the span $(\{J_i^A\}_i)$ can be written as linear combinations of the elements of a set $\{J_i'^A\}_i$ which correspond to TP maps (but not necessarily CP maps) and form a basis for the space spanned by quantum channels. Finally, it is also necessary that these operators can be themselves expressed as a tensor power of a TP map, because they play the role of the inputs of the quantum switch, which takes only a single call of each input. That is, they appear on both sides of the constraint $C_s^*[(J_i'^A)^{\otimes k_A} \otimes (J_j'^B)^{\otimes k_B}] = p\,S^*(J_i'^A \otimes J_j'^B)$.

We now construct a convenient basis for the linear space spanned by the set of $k$ copies of any quantum channel, which will be used in the computer-assisted proofs presented subsequently.

First, note that an arbitrary self-adjoint two-qubit operator $M \in \mathcal{L}(\mathcal{H}^I \otimes \mathcal{H}^O) \cong \mathcal{L}(\mathbb{C}^2 \otimes \mathbb{C}^2)$ can always be written as

$$
\begin{aligned}
M = \ &\lambda\,\mathbb{1} \otimes \mathbb{1} \\
&+ \sum_i \alpha_i \sigma_i \otimes \mathbb{1} + \sum_j \beta_j\,\mathbb{1} \otimes \sigma_j + \sum_{ij} \gamma_{ij} \sigma_i \otimes \sigma_j,
\end{aligned}
\tag{29}
$$

where all $\sigma_i \in \{X, Y, Z\}$ are Pauli operators, and $\lambda, \alpha_i, \beta_j, \gamma_{ij} \in \mathbb{R}$. Since operators $M$ that correspond to a TP map satisfy $\text{tr}_O(M) = \mathbb{1}^I$, in this case one has that $\lambda = 1/2$ and $\alpha_i = 0$. This implies that the dimension of the linear space spanned by the set of qubit quantum channels is $3 + 9 + 1 = 13$. One can then construct a convenient basis for the subspace spanned by qubit quantum channels, given by

$$
\begin{aligned}
\mathcal{B}_1 := \ &\Big\{ \mathbb{1} \otimes \tfrac{1}{2}, \\
&\mathbb{1} \otimes \tfrac{1}{2} + \mathbb{1} \otimes \sigma_i, \\
&\mathbb{1} \otimes \tfrac{1}{2} + \sigma_j \otimes \sigma_k \Big\}_{i,j,k}.
\end{aligned}
\tag{30}
$$

Notice that all elements of the basis set $\mathcal{B}_1$ correspond to TP maps, even if they are not necessarily positive semidefinite.

We now consider the linear space spanned by the set of $k$ copies of any arbitrary qubit channel, i.e., the span of the set of all $J^{\otimes k}$ such that $\text{tr}_O(J) = \mathbb{1}^I$. If a linear subspace $\mathcal{V}$ has dimension $d_\mathcal{V}$, then the dimension of the space span $(\{J^{\otimes k} | J \in \mathcal{V}\})$ is given by $\binom{d_\mathcal{V} - 1 + k}{k}$[28]. By setting $d_\mathcal{V} = 13$ and $k = 2$, we see that the space spanned by the set of two

## BOX 1

# Algorithm for computer-assisted proofs

1. Construct symbolic non-floating-point operators $\Gamma^{sym}$ and $R_{ij}^{sym}$ by truncating $\Gamma^{float}$ and $R_{ij}^{float}$ to obtain symbolic operators expressed only in terms of rational numbers.
   This allows us to work with fractions and to avoid numerical imprecision.

2. Force the operators $\Gamma^{sym}$ and $R_{ij}^{sym}$ to be self-adjoint by making use of the expression $(M + M^\dagger)/2$, which is self-adjoint for any $M$.
   This ensures that we are dealing with self-adjoint operators.

3. Evaluate $t^{sym} := \sum_{i,j} \mathrm{tr}[R_{ij}^{sym}(S^\star(J_i^A \otimes J_j^B))]$, where $S, J_i^A$, and $J_j^B$ are also symbolic operators. Define $R_{ij}^{OK} := R_{ij}^{sym}/t^{sym}$ for all $i, j$.
   Here, we use the basis with rational coefficients constructed earlier to ensure that the operators $R_{ij}^{OK}$ satisfy the SDP equality constraint exactly.

4. Project $\Gamma^{sym}$ onto the appropriate subspace to obtain $\overline{\mathbb{P}}(\Gamma^{sym})$.
   This ensures that the resulting operator $\overline{\mathbb{P}}(\Gamma^{sym})$ is in the correct linear subspace.

5. Find $\eta \in \mathbb{R}$ such that $\Gamma^{OK} := \overline{\mathbb{P}}(\Gamma^{sym}) + \eta \mathbb{1} \geq 0$ and $\Gamma^{OK} - \sum_{i,j} R_{ij}^{OK} \otimes (J_i^{A \otimes k_A} \otimes J_j^{B \otimes k_B})^T \geq 0$.
   This ensures the positivity constraints in the dual SDP (28) are satisfied.

6. Output the quantity $\mathrm{tr}(\Gamma^{OK})/d_{c_i} d_{t_i} d_{A_O}^{k_A} d_{B_O}^{k_B}$, which is a rigorous upper bound of the primal problem.
   All equality and inequality constraints hold without relying on floating-point precision, and the operators $\Gamma^{OK}$ and $R_{ij}^{OK}$ yield a proof certificate that
   $p \leq \mathrm{tr}(\Gamma^{OK})/d_{c_i} d_{t_i} d_{A_O}^{k_A} d_{B_O}^{k_B}$.

identical copies of qubit channels has dimension 91. Hence, in order to find a basis for two identical copies of qubit channels, it is enough to exhibit a set of 91 operators $J_i$, all respecting $\mathrm{tr}_O(J_i) = \mathbb{1}^I$, such that the set $\{J_i^{\otimes 2}\}_{i=1}^{91}$ is composed of linearly independent operators.

Let $\rho_i \in \{\mathbb{1} \otimes \frac{1}{2}, 2|\phi^+\rangle\langle\phi^+|, \mathbb{1} \otimes |0\rangle\langle 0|, \mathbb{1} \otimes |+\rangle\langle +|, \mathbb{1} \otimes |+_Y\rangle\langle +_Y|\}$, where $|\phi^\pm\rangle := \frac{1}{\sqrt{2}}(|00\rangle \pm |11\rangle)$ are maximally entangled two-qubit states and $|\pm_Y\rangle := \frac{1}{\sqrt{2}}(|0\rangle \pm i|1\rangle)$). Then, let $\sigma_i^I \in \{\mathbb{1}, X, Y, Z\}$ and $\sigma_i^O \in \{X, Y, Z\}$. We define the set of operators

$$
\begin{aligned}
\mathcal{B}_2 := \Big\{ & \left(\mathbb{1} \otimes \tfrac{1}{2}\right)^{\otimes 2}, \\
& \left(\rho_i \pm \sigma_j^I \otimes \sigma_k^O\right)^{\otimes 2}, \\
& \left(\rho_l + \sigma_m^I \otimes \sigma_n^O + \sigma_p^I \otimes \sigma_q^O\right)^{\otimes 2} \Big\}_{i,j,k,l,m,n,p,q},
\end{aligned}
\tag{31}
$$

which can be shown to contain a subset of 91 linearly independent operators by standard computational methods. Hence, the set $\mathcal{B}_2$ forms an overcomplete basis, from which one can obtain a standard basis (containing only 91 operators) by discarding any operators that are not linearly independent. This set of 91 operators forms a basis for the linear subspace spanned by two identical copies of any arbitrary qubit channel.

For the case of $k = 3$, we begin by calculating the dimension of the space spanned by three identical copies of arbitrary qubit channels, by setting $d_\gamma = 13$ and $k = 3$, obtaining that the dimension of the relevant space is 455. Again, in order to find a basis for three identical copies of qubit channels, it is enough to exhibit 455 operators $J_i$ respecting $\mathrm{tr}_O(J_i) = \mathbb{1}^I$, such that the set $\{J_i^{\otimes 3}\}_{i=1}^{455}$ is composed of linearly independent operators.

Let $\rho_i \in \{\mathbb{1} \otimes \frac{1}{2}, 2|\phi^+\rangle\langle\phi^+|, 2|\phi^-\rangle\langle\phi^-|, 2|\psi^+\rangle\langle\psi^+|, 2|\psi^-\rangle\langle\psi^-|, \mathbb{1} \otimes |0\rangle\langle 0|, \mathbb{1} \otimes |1\rangle\langle 1|, \mathbb{1} \otimes |+\rangle\langle +|, \mathbb{1} \otimes |-\rangle\langle -|, \mathbb{1} \otimes |+_Y\rangle\langle +_Y|, \mathbb{1} \otimes |-_i\rangle\langle -_i|\}$, where $|\psi^\pm\rangle := \frac{1}{\sqrt{2}}(|01\rangle \pm |10\rangle)$ are maximally entangled two-qubit states and $|-\rangle := \frac{1}{\sqrt{2}}(|0\rangle - |1\rangle)$. Then, let $\sigma_i^I \in \{\mathbb{1}, X, Y, Z, X+Z, X+Y\}$ and

$\sigma_i^O \in \{X, Y, Z, X+Z, X+Y\}$. We define the set of operators

$$
\mathcal{B}_3 := \left\{ \left(\rho_i \pm \sigma_j^I \otimes \sigma_k^O + \sigma_l^I \otimes \sigma_m^O\right)^{\otimes 3} \right\}_{i,j,k,l,m},
\tag{32}
$$

which can be shown to contain a subset of 455 linearly independent operators by standard computational methods. This set of 455 operators forms a basis for the linear subspace spanned by three identical copies of arbitrary qubit channels.

In general, one can always construct a basis for the space spanned by $k \in \mathbb{N}$ copies of arbitrary qubit channels by finding coefficients $\alpha_{i|l}, \gamma_{ij|l} \in \mathbb{R}$, such that the set

$$
\left\{ \left( \mathbb{1} \otimes \tfrac{1}{2} + \sum_i \alpha_{i|l} \mathbb{1} \otimes \sigma_i + \sum_{ij} \gamma_{ij|l} \sigma_i \otimes \sigma_j \right)^{\otimes k} \right\}_l
\tag{33}
$$

contains $\binom{13-1+k}{k}$ linearly independent operators. One simple way to find such coefficients is simply to choose them at random.

To evaluate the SDPs 27 and (28) when the inputs form a basis for the space spanned by $k_A$ copies of channel $A$ and $k_B$ copies of channel $B$, the overall number of input pairs of channels are: for $(k_A, k_B) = (1, 1)$, $N_A N_B = 13 \cdot 13 = 169$; for $(k_A, k_B) = (2, 1)$, $N_A N_B = 91 \cdot 13 = 1183$; for $(k_A, k_B) = (3, 1)$, $N_A N_B = 455 \cdot 13 = 5915$; and finally for $(k_A, k_B) = (2, 2)$, $N_A N_B = 91 \cdot 91 = 8281$, making these SDPs very computationally demanding.

## Computer-assisted proofs

While floating-point arithmetic provides an efficient and powerful numerical method to treat real numbers, it suffers from some fundamentally unavoidable issues. For instance, addition and multiplication of floats is not associative and equality and inequality constraints are not satisfied exactly, but only up to some numerical precision. We now show how efficient numerical solvers that make use of floating-point arithmetic can be used to obtain a rigorous upper bound on the maximal success probability of simulating the quantum switch, hence leading to a bona-fide computer-assisted proof. Our methods are based on ref. 10.

The duality aspects of semidefinite programming[53] ensure that any feasible point of the dual problem, presented in SDP (28), yields an upper bound for the solution of the maximisation problem in the primal SDP 27. That is, any set of operators $\{R_{ij}\}_{ij}$ and $\Gamma$ that respects $\sum_{i,j} \text{tr}[R_{ij}(S^*(J_i^A \otimes J_j^B))] = 1$, $\Gamma - \sum_{i,j} R_{ij} \otimes [(J_i^A)^{\otimes k_A} \otimes (J_j^B)^{\otimes k_B}] \geq 0$, $\Gamma \geq 0$, and $\overline{\mathbb{P}}(\Gamma) = \Gamma$, implies that the probability of simulating the quantum switch is necessarily $p \leq \text{tr}(\Gamma)/d_{c_I} d_{t_I} d_{A_O}^{k_A} d_{B_O}^{k_B}$. Standard numerical SDP solvers can be used to find a floating-point solution for the dual problem in SDP (28), and to provide explicit operators $\{R_{ij}^{\text{float}}\}_{ij}$ and $\Gamma^{\text{float}}$ that approximately satisfy the SDP constraints. We now show how the operators $\{R_{ij}^{\text{float}}\}_{ij}$ and $\Gamma^{\text{float}}$ can be used as a good initial ansatz to construct symbolic operators $R_{ij}^{\text{OK}}$ and $\Gamma^{\text{OK}}$, which are not stored as floating-point variables and which satisfy the SDP constraints exactly, ensuring that $\text{tr}(\Gamma^{\text{OK}})/d_{c_I} d_{t_I} d_{A_O}^{k_A} d_{B_O}^{k_B}$ is a legitimate upper bound for the probability of success of simulating the quantum switch. In order to do so, we use the bases presented in Eqs. (30), (31), (32), and (33), which only contain small rational numbers. In Box 1, we present an algorithm to extract a computer-assisted proof from numerical solvers.

The algorithm described in Box 1 deserves two clarifications. First, since the projector $\overline{\mathbb{P}}$ is unital, i.e., $\overline{\mathbb{P}}(\mathbb{1}) = \mathbb{1}$, for any $\eta \in \mathbb{R}$ and any operator $M$, we have that $\overline{\mathbb{P}}(M) + \eta\mathbb{1} = \overline{\mathbb{P}}(M + \eta\mathbb{1})$. Second, checking if an operator is positive semidefinite can be done efficiently through the Cholesky decomposition—more details are provided in Supplementary Note 6.

## Data availability
No data sets were generated or analyzed during the current study.

## Code availability
All code developed for this work is available at our online repository in ref. 55. For numerically evaluating our SDPs, we used the Splitting Conic Solver (SCS)[56,57] and MOSEK[58].

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

## Acknowledgements

We would like to thank Alastair Abbott, Emanuel-Cristian Boghiu, Cyril Branciard, Anne Broadbent, Giulio Chiribella, Arthur Mehta, Simon Milz, Pierre Pocreau, Louis Salvail, Mykola Semenyakin, Jacopo Surace, and Matt Wilson for helpful discussions. We are also grateful to Denis Rosset and Siegfried M. Rump for helpful discussions regarding computer-assisted proofs. We acknowledge funding from the Swiss National Science Foundation (SNSF) through the funding schemes Swiss Post-doctoral Fellowship (project 216979), NCCR SwissMAP (project 182902), and project 192244 (J.B.); the Japan Society for the Promotion of Science (JSPS) Postdoctoral Fellowships for Research in Japan (P.T.); the JSPS KAKENHI Grant Number 23KJ0734, the FoPM, WINGS Program, and the DAIKIN Fellowship Program (S.Y.); the MEXT Quantum Leap Flagship Program (MEXT QLEAP) JPMXS0118069605, JPMXS0120351339, the JSPS KAKENHI Grant Numbers 21H03394 and 23K21643, and IBM Quantum (M.M.); the Japanese-French Laboratory for Informatics (JFLI) for the support on organizing the Japanese-French Quantum Information 2023 workshop (J.B., H.K., M.M., T.O., M.T.Q., P.T., S.Y.). Research at Perimeter Institute is supported in part by the Government of Canada through the Department of Innovation, Science and Economic Development and by the Province of Ontario through the Ministry of Colleges and Universities (H.K.).

## Author contributions

M.M. conceived the main idea and initiated the project. H.K., M.M., T.O., P.T., and S.Y. developed the results related to deterministic exact simulations. These authors contributed equally: H.K., T.O., and S.Y. J.B. and M.T.Q. developed the results related to probabilistic and approximate simulations. Code: J.B. developed the SDP-related code; M.T.Q. developed the code for the computer-assisted proofs. All authors contributed to discussions, refining of the proofs, and writing the paper.

## Competing interests

The authors declare no competing interests.
