## [Transparent Peer Review file · Nature Communications]

Simulating the quantum switch with quantum circuits is computationally hard

Corresponding Author: Professor Marco Túlio Quintino

A version of this paper was originally rejected for publication by Nature Communications, however that decision was reconsidered after appeal by the authors.

Version 1:

Reviewer comments:

Reviewer #1

(Remarks to the Author)

This paper studies the question of simulation of higher-order quantum transformations (or transformations of transformations) with indefinite causal order using the input operations multiple times in a definite causal order. If we have access to the higher-order transformation, we can achieve the task by using the inputs once, but if we are restricted to use only causally definite higher-order transformations, achieving the task may require using the input operations many times, or may be impossible. The number of uses of the black-box inputs that we need in order to achieve the simulation defines a specific notion of complexity. The question can be defined in different variants. For instance, we may consider perfect, approximate, or probabilistic simulation, or we may be given inputs known to belong to a specific subset of all possible inputs, we may be constrained to use some of the inputs only a fixed number of times, etc.

The main focus of the paper is this type of complexity of simulating the causally non-separable process quantum switch. It has been known that if the input operations are unitary, this can be achieved by calling one of the operations once and the other one twice, and it is obvious from the circuit that the same works when just one of the operations is unitary. What the authors show in this paper is that if the input operations are arbitrary n -qubit quantum channels A and B , and if we consider causal simulations in which B is used once, then A has to be used at least 2^n times for perfect simulation. In the case of approximate and probabilistic simulation, certain results for different scenarios are obtained indicating that the task is still difficult, although no general bound on the complexity as a function of the error is given. Other more specific results are also shown, such as the impossibility of simulating the quantum switch using 2 copies of A and 2 copies of B . It is conjectured that an exponential cost would remain even when we allow both A and B to be used multiple times.

I think that this work is a valuable contribution to the efforts to understanding the information processing power of causally indefinite processes versus processes with definite causal order. It should be of interest to researchers working on higher-order transformations and processes with indefinite causal order. It is well executed and clearly written. However, although I think the results are technically nontrivial, I don't find the message surprising or offering a clearly important conceptual breakthrough. What to me is the most important conceptual contribution of this work is formulating the task to be about implementing the output of the higher-order transformation, as opposed to achieving some specific communication or computation task where the output is given by a classical variable. This is suggested to be an instance of a "computational" task in a generalized sense in which the inputs and outputs are not classical variables but quantum operations. But this notion of "computation" is quite different from the standard notion and the link between the two concepts and the interest in this generalised notion is not clear. The authors also suggest that the result could be important for experimental implementations, where there have been debates about whether certain experiments use the input operations of the quantum switch once or twice. I think it is useful to know that even two uses of each operation, when both operations are nonunitary, cannot simulate the quantum switch. But the essence of these debates is about whether what is happening in the experiments is really a higher-order transformation of the relevant black-box operations. I think this question cannot be settled by the fact that the experiment produces the correct output transformation.

In conclusion, I think that this is a nice piece of work that makes an interesting and useful step in the field, which could inspire further ideas. However, I do not find the results to have clear implications that make a strong case for publication in

this journal.

(Remarks on code availability)

Reviewer #2

(Remarks to the Author)

This paper investigates whether the action of the quantum switch, the canonical example of quantum circuits with quantum control of causal orders, on two local input channels can be simulated deterministically and universally by local quantum channels implemented in a quantum circuit with definite causal orders. While it is known that the answer is positive with an infinite number of local channel calls, the general question for a finite number of resources remains open.

The authors provide a strong negative answer by identifying various cases of finite number of local channel calls for which such simulation is impossible, showing for instance that a one-partite quantum query complexity of the quantum switch is lower bounded by 2^n (the dimension of the local party systems). The robustness of this result is studied via the impossibility of probabilistic and approximate (universal) simulations as well. For various combinations of extra calls, upper bounds are provided using semidefinite programming (SDP). A well-known simulation of the quantum switch in a quantum circuit of type ABA, where A and B are unitary channels, is also generalised, and new cases of deterministic simulations are introduced: in circuits with local qubit channels of types AAAA, and local unitary qubit channels of types AABB and BAAA. Finally, the authors conjecture the impossibility of any simulation with multiple sub-exponential number of calls to both input channels.

This is a highly significant contribution to the field of quantum information theory, and especially within the subfields of higher-order quantum computation and indefinite causal orders. The quantum switch is the canonical example of a physical process with indefinite causal orders, and is known to give various advantages across a broad range of information processing tasks. Understanding the resource requirements for simulating such a process is therefore foundational. The authors extend and strengthen prior findings, discussed in this paper, e.g. that the quantum switch could be simulated by circuits with definite causal orders under an infinite amount of resources, via post-selection, or in the special case of unitary resources. The results are also especially relevant in the context of the experimental implementation debate regarding the quantum switch, emphasizing that experiment certifications using non-unitary local channels would not be subject to query complexity loopholes. This work also paves the way to the study of higher order quantum computation query complexity.

The conclusions and claims are very well supported. The paper is clear, well-written, well referenced, detailed and generally well organised. The robustness of the deterministic simulation no-go theorem is backed up by probabilistic and approximate simulation numerical analysis. The probabilistic simulation analysis is made via rigorous upper-bounds obtained via a computer assisted proof. The method is explained in details in the main text. A furnished SM is provided, and the codes used for numerical results as well. While I have verified the global soundness of the proofs in SM, I did not verify the numerical results. The methodology is perfectly sound, meet the expected standards in the field and is sufficiently detailed for reproducibility.

I did not identify any flaws in the data analysis, interpretation nor conclusion. The authors acknowledge the open problem of whether any finite number of channel calls could allow for a simulation, which is an honest and critical delineation of the study's scope.

The manuscript is already well-crafted and suitable for publication. The following non-mandatory suggestions may improve clarity:

- 89-90: “quantum circuit with classical control of causal orders” appears for the first time. Add eventually the first call of the reference [29] here ?
- 172: Eventually add “a deterministic simulation” ?
- In both introduction and discussion, the “go-theorem” is presented at the end. Why not follow this structure and put the according section after the “no-go theorem” sections ?
- 279, 305, 308, 732, ..., SM: the use of the terminology “bipartite channel” (and “one-party”) might be slightly confusing, as, in the process matrix formalism, a formalism typically used to study indefinite causal orders, a party is designated as a local agent implementing a quantum instrument. In this formalism nomenclature, the “bipartite channels” would simply be interpreted as one-partite operation with auxiliary systems (the tilde additional spaces). An explanatory note / comment might be useful here.
- 387: add eventually a reference to Abbott, Mhalla & Pocreau, Phys. Rev. Research 6, L032020 (2024) (<https://arxiv.org/abs/2307.10285>)
- Figure 4: An explicit comment on the epsilon values thresholds for which the quantum switch start being perfectly approximated could be added.
- 635-636: Is there any intuition/interpretation behind this result ?

- Use eventually more systematically the terminology “quantum combs” or “QCFO” (quantum circuit with fixed causal order) to designate “standard quantum circuit” ?
- SM, p.11 “The proof is based upon a series of lemmas...” eventually explain briefly also here the uses of lemma 1, 6 and 7 ?
- SM, p.14 Eq. C17, missing “=1” under the first sum. Eq. C18, find eventually a clearer way to write the sum index constraining the i s and j s.
- SM, p.15 “Therefore...”, add “from Lemma 1” ?

To conclude, this is a technically sound, insightful, and impactful study. It makes significant progress in a foundational question of quantum information theory. The results will likely shape future theoretical and experimental work on indefinite causal order, higher-order transformations, and quantum resource theories.

(Remarks on code availability)

Reviewer #3

(Remarks to the Author)

In the manuscript, the authors propose the question whether the quantum switch--both its standard form and an extended form, that acts only on part of the input channels--can be deterministically simulated using processes with classically allowed causal ordering, and answer it in the negative for a limited amount of resources, aside from very particular cases where the simulation is possible.

The results include the impossibility of perfectly simulating the extended quantum switch for general n -qubit channels as inputs with fewer than an exponential amount of repetitions of one of the channels, and hence a notion of an exponential advantage in query complexity by having access to a quantum switch, formalized in the introduction of the notion of one-sided quantum query complexity.

Also included is a positive result in the generalization of the simulability of the quantum switch to its extended version, when at least one of the inputs is a unitary channel, using only two repetitions of the unitary channel; the possibility (numerically obtained) of performing restricted simulations when inputs are identical qubit channels, when at least four uses of the channel are allowed in the simulation; and in the negative, computer-aided proofs of nonsimulability in the case of two uses of both input channels, besides other cases.

The paper is well-written, clear, and technically sound. All proofs seem correct and the numerical methods presented are straightforward and valid. The results are interesting and significant to warrant publication on Nature Communications.

Although not impacting the assessment of the article, some remarks are in order: From the “go theorem” for the simulability of the extended quantum switch using unitary channels, there seems to be possible the simulability of the standard quantum switch on general channels due to the possibility of realizing it on their Stinespring dilation. The impossibility of performing such an adaptation is addressed in the supplementary information, but the authors should consider including this discussion in the main article, as it is a natural question to arise after seeing Theorem 1.

Then, on the numerical results that reveal different performances in the restricted simulation of the quantum switch when different orders of the gates are used, do the authors have an insight for why is that the case? If so, a comment on it would be interesting.

On minor issues, some typos identified are listed below:

On line 922 - It should be “contradiction” instead of “contraction”.

On line 1038 - The set should be defined over all J in V , without the copies on the right-hand side.

(Remarks on code availability)

Version 2:

Reviewer comments:

Reviewer #1

(Remarks to the Author)

I have been convinced by the reply of the authors and am satisfied by the introduced changes I am happy to recommend the paper for publication.

(Remarks on code availability)

I have been convinced by the reply of the authors and am satisfied by the introduced changes I am happy to recommend the paper for publication.

Reviewer #3

(Remarks to the Author)

All points raised by the referees have been adequately addressed by the authors. The results are relevant to the field of higher-order processes and have a foundational significance in establishing an even stronger separation from processes with classically allowed causal orders. Even if an expected result, it hasn't been clearly proven before and the present work introduces a significant step in this direction. As such, this work will likely be a reference in subsequent works that tackle the subject of higher order quantum computation and the potential associated advantages. I maintain my previous recommendation of publication in the journal.

(Remarks on code availability)

Reply to Referees

We thank the reviewers for the time and effort dedicated to providing feedback on our manuscript. Their comments and suggestions have helped us strengthen our work and we hope that it now meets the standards for publication in Nature Communications. Please find below a list of the changes in the resubmitted manuscript followed by a detailed reply to each individual point that was raised.

Additionally from the changes that were motivated by the feedback of the referees, we also shortened some parts of the text to comply with the length guidelines of Nature Communications. These changes did not impact the content of the manuscript and helped us improve the presentation.

The changes that were made in the main text following the advice of the referees are highlighted in yellow. The changes that were made to reduce the length text are coloured in green.

List of changes:

- Removed mentions of quantum combs from the introduction; kept the intuitive terminology “quantum circuits with fixed or classically-controlled causal order” in the non-technical parts of the manuscript and used the more precise terms “quantum comb” and “QC-CC” in technical parts of the manuscript (various highlighted places).
- Added an earlier reference to [27] (in line 85) and to [18] (in line 352).
- Added “deterministic and exact” (in line 145).
- Rephrased the explanation of the terms “quantum comb” and “QC-CC” (sentences starting on 155 and 159).
- Added sentence about the notion of quantum computation with quantum inputs and outputs (line 175).
- Added interpretation of bipartite channels (in line 232).
- Added discussion on the Stinespring dilation on the main text (in line 281).
- Added explicit thresholds for epsilon, extracted from the plots in Fig. 4, to the main text (in lines 562 and 566).
- Clarified and rephrased parts of the discussion on the implications of our results to experimental analysis (in paragraph that starts at line 659).
- Fixed typos.
- Updated Ref. [13]; some references were removed as a consequence of the discussion about arbitrary transformations of unitaries being removed from the Discussion to shorten the main text.
- In the SI:
 - Explicit mention of certain lemmas when they are used in the proofs.
 - Fixed typo in Eq. (C14).
 - The table of contents includes subsections of all lemmas.
 - Simplified notation in the statements of the lemmas; small improvements in the presentation of their proofs.
 - General definition of QC-CCs moved to before the statement of Lemma 5, instead of in the lemma environment.

Replies:

Reviewer #1 (Remarks to the Author):

This paper studies the question of simulation of higher-order quantum transformations (or transformations of transformations) with indefinite casual order using the input operations

multiple times in a definite casual order. If we have access to the higher-order transformation, we can achieve the task by using the inputs once, but if we are restricted to use only causally definite higher-order transformations, achieving the task may require using the input operations many times, or may be impossible. The number of uses of the black-box inputs that we need in order to achieve the simulation defines a specific notion of complexity. The question can be defined in different variants. For instance, we may consider perfect, approximate, or probabilistic simulation, or we may be given inputs known to belong to a specific subset of all possible inputs, we may be constrained to use some of the inputs only a fixed number of times, etc.

The main focus of the paper is this type of complexity of simulating the causally non-separable process quantum switch. It has been known that if the input operations are unitary, this can be achieved by calling one of the operations once and the other one twice, and it is obvious from the circuit that the same works when just one of the operations is unitary. What the authors show in this paper is that if the input operations are arbitrary n -qubit quantum channels A and B , and if we consider causal simulations in which B is used once, then A has to be used at least 2^n times for perfect simulation.

In the case of approximate and probabilistic simulation, certain results for different scenarios are obtained indicating that the task is still difficult, although no general bound on the complexity as a function of the error is given. Other more specific results are also shown, such as the impossibility of simulating the quantum switch using 2 copies of A and 2 copies of B . It is conjectured that an exponential cost would remain even when we allow both A and B to be used multiple times.

We thank the referee for their diligent review and summarizing our work. We believe the referee understood our main result, but in order to be very precise, we remark that our result implies that with at most 2^n calls of A , a perfect simulation is not possible. While this indeed implies that, if it were possible to perfectly simulate the quantum switch, at least 2^n calls of A would be needed, the possibility of exact simulation itself remains an open question. As we remark in our discussion, it could be that even any finite number of calls does not suffice.

I think that this work is a valuable contribution to the efforts to understanding the information processing power of causally indefinite processes versus processes with definite casual order. It should be of interest to researchers working on higher-order transformations and processes with indefinite casual order. It is well executed and clearly written. However, although I think the results are technically nontrivial, I don't find the message surprising or offering a clearly important conceptual breakthrough.

We are very grateful to the referee for deeming our work a valuable contribution to the field, and finding it to be well executed and clearly written. Regarding not finding our message surprising, we agree to a certain extent. We initially believed it to be impossible to simulate the switch, at least efficiently, and were not surprised by our results confirming this, going as far as to conjecture them to hold in more generality than we were able to prove. However, we believe that this expectation does not diminish the impact and the highly non-trivial aspect of our work—we have answered a long-standing question that had been open since the very first paper that considered computation with indefinite causal order, Ref. [6], which first appeared on the arXiv in 2009. This paper is widely regarded as seminal, having spawned the entire research program of higher order quantum computation—definitively answering its key open problem will have widespread repercussions.

At the same time, while not entirely surprising to us, there exists a large part of the community working in higher-order transformations, as well as non-specialists, that might have expected a general

simulation of the quantum switch to be easy (or at least possible). This is in part because the simulation of the quantum switch for unitary channels is so simple (and it might not be obvious that it does not straightforwardly generalize) or because they do not typically consider non-unitary evolutions. We expect that our results could indeed be surprising for these readers and provide a solid foundation regarding the many intricacies involved.

What to me is the most important conceptual contribution of this work is formulating the task to be about implementing the output of the higher-order transformation, as opposed to achieving some specific communication or computation task where the output is given by a classical variable. This is suggested to be an instance of a "computational" task in a generalized sense in which the inputs and outputs are not classical variables but quantum operations. But this notion of "computation" is quite different from the standard notion and the link between the two concepts and the interest in this generalised notion is not clear.

We thank the referee for acknowledging our important conceptual contribution in terms of the task we formulate.

As pointed out by the referee, our manuscript considers a task where the inputs and outputs are quantum objects. While it is true that a significant part of the field of quantum computing has studied quantum computational models with the goal of solving classical computing problems, where the inputs and outputs are classical but processed through a quantum transformation, we argue that our notion of quantum-to-quantum computation has also been extensively studied by the quantum computing community since at least the late 90s, and as such is not a generalization that we propose, but a solid and well-established field.

A pioneer of such examples is the swap test [*SIAM Journal on Computing*. **26** (5): 1541–1557, (1997)], published in 1997 and rediscovered in 2001 under the name quantum fingerprint [*Phys. Rev. Lett.* **87**, 167902 (2001)]. In this task, one is given a pair of quantum states as inputs and aims to estimate their overlap. Crucially, the input states can be fully arbitrary and the task has no classical analogue; nonetheless, it is today recognised as a basic building block for quantum computing. In the literature, such computational tasks with quantum inputs are sometimes phrased as “quantum testing of quantum properties”, and recognised as a valid notion of quantum computation [see, for instance, the survey article *Theory of Computing, Graduate Surveys 7* (2016)]. Additionally, canonical quantum algorithms such as Deutsch, Simon, and Grover implicitly consider quantum inputs in the form of quantum oracles, much in the same way as we consider quantum channels as inputs in higher-order quantum computing. Furthermore, the study of Hamiltonian transformations, molecular simulation, and quantum chemistry, which use quantum computing methods to study intrinsic quantum problems, with quantum inputs as well as quantum outputs, has also long been carried out in the field.

While we agree that our work does not necessarily make a connection between the notion of quantum computing for quantum problems (our focus) and the notion of quantum computing for classical problems (the notion that the referee mentions), we do not see this as a shortcoming, but rather as a different scope pertaining to a related and equally interesting field.

We have added a sentence to our manuscript putting our notion of computation into context with the literature.

The authors also suggest that the result could be important for experimental implementations, where there have been debates about whether certain experiments use the input operations of the quantum switch once or twice. I think it is useful to know that even two uses of each operation, when both operations are nonunitary, cannot simulate the quantum switch. But the essence of these debates is about whether what is happening in the experiments is really a

higher-order transformation of the relevant black-box operations. I think this question cannot be settled by the fact that the experiment produces the correct output transformation.

We completely agree with the referee on this point. It is true that our results do not settle the debates concerning the experimental implementations or the quantum switch. At the same time, we remark that settling this question is very much out of the scope of our work. We also completely agree that the debates about the quantum switch cannot be settled by the fact that the experiment produces the correct output transformation, and as such, we do not make any claims of this sort in our work.

However, as correctly pointed out by the referee, our results do have *some* relevant implications for the experimental analysis. We discuss these exact implications in our manuscript, and refrain from discussing other potential experimental loopholes that our work does not contribute toward understanding.

To make all of these important points more clear, we revised our discussions concerning the implication of our results for experimental analysis. For the sake of the length of the main text, we removed the discussion about experiments from the Introduction and focused it all in the Discussion. There, we rephrased parts of the discussion that we felt could have led to a potential misunderstanding of our claims and stated our points more clearly.

Finally, we remark that, while only a contributing step in this debate, the pertinent results we show—namely that (1) any causal simulation model that takes two independent calls of black-box quantum channels cannot simulate the quantum switch deterministically and exactly, and (2) it is impossible to transform two independent calls of black-box quantum channels into “two correlated uses” of a quantum channel, a resource which is known to allow for a causal simulation—address important questions that went unanswered for over a decade before our work. These results are also of independent theoretical interest.

In conclusion, I think that this is a nice piece of work that makes an interesting and useful step in the field, which could inspire further ideas. However, I do not find the results to have clear implications that make a strong case for publication in this journal.

We thank the referee for their comments and for helping us clarify the discussions we present in our manuscript.

While our work still leaves room for relevant open questions, we have made significant progress on one of the most important open problems in higher-order quantum computing. Namely, we have shown the first exponential computational advantage of indefinite causal order and closed the question of the simulability of the quantum switch in arguably the most relevant practical scenario, that of two calls to each input channel. Not only that, but our results are incredibly robust: our no-go results are applicable under minimal assumptions (convex combinations of unitary channels in Theorem 2 and qubit channels in Theorem 3) and consider also restricted, probabilistic, and even approximate simulation scenarios. Our technical methods are solid and rigorous, going far beyond the standard practices for treating numerical results in the literature and offering a diligent analytical proof of a highly non-trivial result. We have furthermore put an immense effort into our writing to present our results in an accessible, interesting, yet still precise way.

We hope to have made the relevance and impact of our work more clear in the resubmission, and would like to kindly ask the referee to reconsider their recommendation.

Reviewer #2 (Remarks to the Author):

This paper investigates whether the action of the quantum switch, the canonical example of quantum circuits with quantum control of causal orders, on two local input channels can be simulated deterministically and universally by local quantum channels implemented in a quantum circuit with definite causal orders. While it is known that the answer is positive with an infinite number of local channel calls, the general question for a finite number of resources remains open.

The authors provide a strong negative answer by identifying various cases of finite number of local channel calls for which such simulation is impossible, showing for instance that a one-partite quantum query complexity of the quantum switch is lower bounded by 2^n (the dimension of the local party systems). The robustness of this result is studied via the impossibility of probabilistic and approximate (universal) simulations as well. For various combinations of extra calls, upper bounds are provided using semidefinite programming (SDP). A well-known simulation of the quantum switch in a quantum circuit of type ABA, where A and B are unitary channels, is also generalised, and new cases of deterministic simulations are introduced: in circuits with local qubit channels of types AAAA, and local unitary qubit channels of types AABB and BAAA. Finally, the authors conjecture the impossibility of any simulation with multiple sub-exponential number of calls to both input channels.

We thank the referee for this careful summary of our results.

This is a highly significant contribution to the field of quantum information theory, and especially within the subfields of higher-order quantum computation and indefinite causal orders. The quantum switch is the canonical example of a physical process with indefinite causal orders, and is known to give various advantages across a broad range of information processing tasks. Understanding the resource requirements for simulating such a process is therefore foundational. The authors extend and strengthen prior findings, discussed in this paper, e.g. that the quantum switch could be simulated by circuits with definite causal orders under an infinite amount of resources, via post-selection, or in the special case of unitary resources. The results are also especially relevant in the context of the experimental implementation debate regarding the quantum switch, emphasizing that experiment certifications using non-unitary local channels would not be subject to query complexity loopholes. This work also paves the way to the study of higher order quantum computation query complexity.

We thank the referee for their acknowledgement of the significance of the problem we tackle in our work and the impact of our results.

The conclusions and claims are very well supported. The paper is clear, well-written, well referenced, detailed and generally well organised. The robustness of the deterministic simulation no-go theorem is backed up by probabilistic and approximate simulation numerical analysis. The probabilistic simulation analysis is made via rigorous upper-bounds obtained via a computer assisted proof. The method is explained in details in the main text. A furnished SM is provided, and the codes used for numerical results as well. While I have verified the global soundness of the proofs in SM, I did not verify the numerical results. The methodology is perfectly sound, meet the expected standards in the field and is sufficiently detailed for reproducibility.

We thank the referee for acknowledging the mathematical rigour of our methods.

I did not identify any flaws in the data analysis, interpretation nor conclusion. The authors acknowledge the open problem of whether any finite number of channel calls could allow for a simulation, which is an honest and critical delineation of the study's scope.

The manuscript is already well-crafted and suitable for publication. The following non-mandatory suggestions may improve clarity:

We are very grateful to the referee for their thorough reading of our manuscript and SI, for their constructive comments, and for considering that our manuscript is suitable for publication.

• 89-90: "quantum circuit with classical control of causal orders" appears for the first time. Add eventually the first call of the reference [29] here ?

Indeed, we agree that this is the most appropriate point for this reference (formerly Ref. [29], now Ref. [27]) to be cited for the first time. We have now added it and thank the referee for pointing it out.

• 172: Eventually add "a deterministic simulation" ?

Added. We took the opportunity to write "deterministic and exact simulation", to be more precise.

• In both introduction and discussion, the "go-theorem" is presented at the end. Why not follow this structure and put the according section after the "no-go theorem" sections ?

We understand the referee's point, however, let us clarify our reasoning for having chosen to present the results in this order. In the main text, the go theorem comes before the no-go theorem because it is a generalization of the circuit from Ref. [6], which is the only partial simulation result concerning the quantum switch in the literature previously to our work. Our goal in reproducing the circuit of Ref. [6] here is to help the reader understand, firstly, how the switch acts on unitary channels, and second, how non-trivial the problem of generalizing this result to general non-unitary channels is. We then continue the narrative by immediately showing that this result can be generalized, but only to a certain extent—to bipartite unitary channels A and bipartite general channels B. We then proceed to show that a further generalization is impossible, with our no-go theorems.

In the introduction and discussion, however, we mentioned the go theorem together with the numerical "go" results, which also show the existence of a simulation in different non-universal settings (such as for the quantum switch acting on four identical qubit channels in a restricted scenario), but we did not want to go into more technical details of explaining the particularities of each of these results at this point in the text. At the same time, the no-go theorems are the most crucial and impactful results in the paper, and as such, we chose to highlight them by mentioning them first in the Introduction and Discussion.

We believe that by keeping the original order in the Introduction and Discussion, as well as in the results section, we are highlighting the most crucial results first when they are mentioned in passing, and still allowing the narrative of the paper to flow naturally from one technical result to the next.

• 279, 305, 308, 732, ..., SM: the use of the terminology "bipartite channel" (and "one-party") might be slightly confusing, as, in the process matrix formalism, a formalism typically used to study indefinite causal orders, a party is designated as a local agent implementing a quantum instrument. In this formalism nomenclature, the "bipartite channels" would simply be interpreted as one-partite operation with auxiliary systems (the tilde additional spaces). An

explanatory note / comment might be useful here.

We have added a note explaining this point where bipartite channels first appear in the main text, however, we refer to the extra input/output system as an environment, and not as an auxiliary system.

We remark that we use the nomenclature “bipartite channel” because we see it indeed as a two-party channel: the interpretation given by the referee of this operation corresponding to a one-party channel would imply that the acting party could have access to this “auxiliary” system, when this is in fact not the case. The local party, let us say Alice, only has access to one input and one output of the quantum channel, while the other input/output could be seen as an environment or an extra party that is inaccessible to her. We remark that Fig. 3 was originally included in the manuscript to provide a visual representation of this point. If Alice had access to this other input/output, we would be back to the single-party channel scenario of Fig. 1.

• *387: add eventually a reference to Abbott, Mhalla & Pocreau, Phys. Rev. Research 6, L032020 (2024) (<https://arxiv.org/abs/2307.10285>)*

We included a reference to the paper mentioned by the referee at this point in the text. We remark that this paper is already cited earlier in our manuscript, the first time being in the introduction as Ref. [18], but we agree that it makes sense to cite it here as well.

• *Figure 4: An explicit comment on the epsilon values thresholds for which the quantum switch start being perfectly approximated could be added.*

We thank the referee for this suggestion and have now added the explicit values in the main text.

• *635-636: Is there any intuition/interpretation behind this result ?*

We have a partial understanding: First, notice that the simulation in Fig. 4 is for a quantum switch with a fixed input target and control states (in the area jargon, fixed past). In this case, for $k_A=k_B=1$, QC-CCs are equivalent to a convex combination of standard (fixed-order) quantum circuits. What we observe in Fig. 4(a) is that convex combinations of quantum circuits do not improve the probability of success of simulating the quantum switch, even approximately for some error. The interpretation then is that convex combinations do not bring advantage to this kind of simulation, which, one might argue, could be expected in the exact simulation case, but is somewhat surprising in the approximate simulation case. However, we do not have a clear intuition of why this is the case, nor do we have a general proof of this statement, which is why it is not mentioned in the paper.

• *Use eventually more systematically the terminology "quantum combs" or "QCFO" (quantum circuit with fixed causal order) to designate "standard quantum circuit" ?*

We appreciate the referee’s suggestion to standardise our terminology and made changes in the manuscript to improve it in this regard.

We removed the term “standard quantum circuit”. Now, in non-technical parts of the manuscript, we are using “quantum circuits with fixed or classically-controlled causal order”, because we think this terminology is intuitive, even for non-specialists. This is used in the Introduction, Discussion, captions (except Fig. 4, which contains technical results), and theorem statements.

In more technical parts of the manuscript, we are using the technical jargon “quantum comb and QC-CCs”, which are defined in the beginning of the Results section, where they first appear, as quantum circuits with open slots that have a fixed and a classically-controlled causal order. This is

needed both to simplify the terminology of the technical part as well as to allow us to be mathematically precise, for example, when referring to the topology of the sets of quantum combs and QC-CCs.

Finally, we opted for quantum combs instead of the acronym QC-FO because the former jargon has been in use in the literature for over 15 years and thought it would be more familiar to specialists.

- SM, p.11 "The proof is based upon a series of lemmas..." eventually explain briefly also here the uses of lemma 1, 6 and 7 ?

We included a mention of the role of these lemmas in the proof of Theorem 2 in the SI. Lemmas 1, 6, and 7 concern general results in linear algebra and are "auxiliary" lemmas in the proof of the "main" lemmas, which are Lemmas 2-5. The latter are results concerning more specifically the quantum switch simulation problem. We included Lemma 6 and 7 in the table of contents of the SI, which was initially missing, for easier navigation of the SI.

- SM, p.14 Eq. C17, missing "=1" under the first sum. Eq. C18, find eventually a clearer way to write the sum index constraining the i s and j s.

Thank you for pointing this out, it has been fixed.

- SM, p.15 "Therefore...", add "from Lemma 1" ?

We have included this clarification.

To conclude, this is a technically sound, insightful, and impactful study. It makes significant progress in a foundational question of quantum information theory. The results will likely shape future theoretical and experimental work on indefinite causal order, higher-order transformations, and quantum resource theories.

We thank the referee again for their positive assessment of the quality and expected impact of our work, and for the helpful suggestions which helped us improve our manuscript.

Reviewer #3 (Remarks to the Author):

In the manuscript, the authors propose the question whether the quantum switch--both its standard form and an extended form, that acts only on part of the input channels--can be deterministically simulated using processes with classically allowed causal ordering, and answer it in the negative for a limited amount of resources, aside from very particular cases where the simulation is possible.

The results include the impossibility of perfectly simulating the extended quantum switch for general n -qubit channels as inputs with fewer than an exponential amount of repetitions of one of the channels, and hence a notion of an exponential advantage in query complexity by having access to a quantum switch, formalized in the introduction of the notion of one-sided quantum query complexity.

Also included is a positive result in the generalization of the simulability of the quantum switch to its extended version, when at least one of the inputs is a unitary channel, using only two repetitions of the unitary channel; the possibility (numerically obtained) of performing restricted simulations when inputs are identical qubit channels, when at least four uses of the

channel are allowed in the simulation; and in the negative, computer-aided proofs of nonsimulability in the case of two uses of both input channels, besides other cases.

We thank the referee for this careful summary of our results.

The paper is well-written, clear, and technically sound. All proofs seem correct and the numerical methods presented are straightforward and valid. The results are interesting and significant to warrant publication on Nature Communications.

We are very grateful to the referee for their assessment of the quality and soundness of our work, and for finding our manuscript suitable for publication.

Although not impacting the assessment of the article, some remarks are in order: From the "go theorem" for the simulability of the extended quantum switch using unitary channels, there seems to be possible the simulability of the standard quantum switch on general channels due to the possibility of realizing it on their Stinespring dilation. The impossibility of performing such an adaptation is addressed in the supplementary information, but the authors should consider including this discussion in the main article, as it is a natural question to arise after seeing Theorem 1.

Indeed, we included a more detailed comment about this important point in the main text. We agree that the impossibility of simulating the quantum switch using the circuit in Theorem 1, even despite the Stinespring dilation theorem, is a very relevant point, which is why we dedicated so much effort in detailing this in the SI. We now also discuss it explicitly in the main text.

Then, on the numerical results that reveal different performances in the restricted simulation of the quantum switch when different orders of the gates are used, do the authors have an insight for why is that the case? If so, a comment on it would be interesting.

First, let us remark that the upper bounds on the probability of success that we prove and present in Table I are, strictly speaking, not tight (seen from the fact that by construction they are always rational numbers). Hence, they do not necessarily imply a hierarchy between the different fixed orders of simulation. Our computer-assisted proof method allows for these bounds to be tightened as much as desired, given sufficient computational resources. For instance, from our numerical results (all code is available) we know that the $(k_A, k_B) \in \{(1,1), (2,1)\}$ cases in the table are very much close to the optimal result, and indeed the hierarchy of orders for the case $(k_A, k_B) = (2,1)$ is as suggested by the upper bounds. Unfortunately, we do not have an insight on why that is the case.

On minor issues, some typos identified are listed below:

On line 922 - It should be "contradiction" instead of "contraction".

Indeed, thank you for pointing this out. In the revised version we are no longer using the word contradiction in this sentence.

On line 1038 - The set should be defined over all J in V , without the copies on the right-hand side.

We thank the referee for noticing this subtle typo. We have corrected it in our revised manuscript.

We thank the referee once more for these comments, which helped us improve our manuscript.